# Experiences of menstruation in high income countries: A systematic review, qualitative evidence synthesis and comparison to low- and middle-income countries

Dani Jennifer Barrington[1,2]*, Hannah Jayne Robinson[2], Emily Wilson[3], Julie Hennegan[4,5]

1 School of Population and Global Health, The University of Western Australia, Crawley, Western Australia, Australia, 2 School of Civil Engineering, University of Leeds, Leeds, West Yorkshire, United Kingdom, 3 Irise International, Sheffield, South Yorkshire, United Kingdom, 4 Maternal, Child and Adolescent Health Program, Burnet Institute, Melbourne, Victoria, Australia, 5 Melbourne School of Population and Global Health, University of Melbourne, Melbourne, Victoria, Australia

* dani.barrington@uwa.edu.au

**Data Availability Statement:** This manuscript made use of secondary data in the form of publications reporting on menstrual experiences in

## Abstract

### Background

There is growing recognition of the importance of menstruation in achieving health, education, and gender equality for all. New policies in high income countries (HICs) have responded to anecdotal evidence that many struggle to meet their menstrual health needs. Qualitative research has explored lived experiences of menstruating in HICs and can contribute to designing intervention approaches. To inform the growing policy attention to support people who menstruate, here we review and synthesise the existing research.

### Methods and findings

Primary, qualitative studies capturing experiences of menstruation in HICs were eligible for inclusion. Systematic database and hand searching identified 11485 records. Following screening and quality appraisal using the EPPI-Centre checklist, 104 studies (120 publications) detailing the menstrual experiences of over 3800 individuals across sixteen countries were included. We used the integrated model of menstrual experiences developed from studies in low- and middle-income countries (LMICs) as a starting framework and deductively and inductively identified antecedents contributing to menstrual experiences; menstrual experiences themselves and impacts of menstrual experiences. Included studies described consistent themes and relationships that fit well with the LMIC integrated model, with modifications to themes and model pathways identified through our analysis. The socio-cultural context heavily shaped menstrual experiences, manifesting in strict behavioural expectations to conceal menstruation and limiting the provision of menstrual materials. Resource limitations contributed to negative experiences, where dissatisfaction with menstrual practices and management environments were expressed along with feelings of disgust if participants felt they failed to manage their menstruation in a discrete, hygienic

high income countries. Table 2 and the Reference list provide the details of all publications included in this systematic review.

**Funding:** In 2019, Hannah Robinson undertook a summer internship with Irise International, with her living expenses supported by Leeds for Life Foundation Funding. Hannah is currently a PhD candidate supported by the Engineering and Physical Sciences Research Council Grant number EP/S022066/1, although this manuscript does not form part of her PhD work. Neither funder has had any role in study design, data collection and analysis, decision to publish, or preparation of the manuscript.

**Competing interests:** The authors have declared that no competing interests exist.

way. Physical menstrual factors such as pain were commonly associated with negative experiences, with mixed experiences of healthcare reported. Across studies participants described negative impacts of their menstrual experience including increased mental burden and detrimental impacts on participation and personal relationships. Positive experiences were more rarely reported, although relationships between cis-women were sometimes strengthened by shared experiences of menstrual bleeding. Included studies reflected a broad range of disciplines and epistemologies. Many aimed to understand the constructed meanings of menstruation, but few were explicitly designed to inform policy or practice. Few studies focused on socioeconomically disadvantaged groups relevant to new policy efforts.

## Conclusions

We developed an integrated model of menstrual experience in HICs which can be used to inform research, policy and practice decisions by emphasising the pathways through which positive and negative menstrual experiences manifest.

## Review protocol registration

The review protocol registration is PROSPERO: CRD42019157618.

## Introduction

At any given moment approximately 10% of the global population is experiencing their menstrual period [1]. Although menstruation has historically been under-researched [2], there is growing attention to its importance in achieving health, education and gender equality for all [3], including through the recent publication of the definition of menstrual heath as "a state of complete physical, mental, and social well-being and not merely the absence of disease or infirmity, in relation to the menstrual cycle" [pg. 2, 4]. In high income countries (HICs), there have been increasing efforts to understand and address menstrual disorders and pain [2], as well as the links between these and negative consequences for employment and education [5]. There is also growing anecdotal evidence that many people who menstruate do not have access to menstrual materials due to financial constraints [referred to as 'period poverty', 6], with reports that adolescents around the world are missing school due to a lack of access to menstrual materials [7, 8] and those experiencing homelessness are using makeshift menstrual materials such as toilet paper [9, 10]. There has been a consequent overwhelming policy response to provide free menstrual materials. For example, in Scotland, free menstrual materials will soon be available for all who want them [11] and in Victoria, Australia, it was announced in 2019 that all government schools will provide free menstrual pads [12]. These multi-million-dollar programmes have rarely been based on robust research, even though several decades of research from a variety of social science disciplines have highlighted the negative constructions of menstruation throughout society, and feminist campaigners have conceptualised the issue as one of gender-based injustice [13].

In 2019 the UK Government announced its campaign to end period poverty and menstrual shame nationally by 2025 and globally by 2030. To do so they established a taskforce of public sector, private sector, not-for-profit and academic institutions and individuals [14]. This initiative and growing pressure for other HIC governments to act has highlighted the need for more

evidence to inform policy development and the opportunity to learn from the rapidly growing body of research and advocacy work on this issue in low- and middle-income countries (LMICs). Population health research across LMIC settings has elucidated a wide range of contributors to menstrual experiences and impacts on health and well-being through a large body of qualitative research. In 2019, this work was brought together through a systematic review and qualitative evidence synthesis which developed an integrated model of menstrual experiences in LMICs [15]. This model has served as a useful framework for understanding menstrual health in LMICs and has helped to inform subsequent research and practice approaches. It is unclear the extent to which this model is applicable in HICs.

To inform the growing policy attention to support people who menstruate in HICs, through this review we identified and synthesised the existing research on menstrual experiences in these countries. We aimed to; 1) collate the existing body of qualitative research on menstrual experiences in HICs and appraise its quality; 2) synthesise this evidence base and develop a model of menstrual experience relevant to HICs, to understand contributing factors, menstrual experiences themselves and the impacts of menstrual experiences on the lives of people who menstruate; and 3) compare findings to the integrated model of menstrual experience developed based on studies in LMICs, in light of differences in the study populations and research topics.

## Methodology

The review protocol is registered on PROSPERO: CRD42019157618 (https://www.crd.york.ac.uk/prospero/display_record.php?ID=CRD42019157618) and is reported according to PRISMA guidance [16, PRISMA Checklist included as S1 Checklist].

### Search strategy and eligibility

The search strategy was designed to capture all qualitative studies, or mixed method studies that included qualitative methods, reporting on experiences of menstruation (Table 1). Searching was undertaken in 9 databases in July 2019 and updated in November 2020 (Applied Social Science Index and Abstracts, Cumulative Index of Nursing and Allied Health Literature (CINAHL), ProQuest Dissertation and theses, Embase, Global Health, MEDLINE, OpenGrey, PsycINFO, Sociological abstracts) (Fig 1).

To supplement database searches, in September 2019 a list of menstrual health organisations and individuals conducting work in HICs was compiled through their registration as a partner of Menstrual Hygiene Day (https://menstrualhygieneday.org/get-involved/partnership/) or a member of the Menstrual Health Hub (https://mhhub.org/community/global-mh-registry/).

**Table 1. Embase search strategy.**

| Search 1: Menstrual | (menstruation or menarche or menstrual period or menstru* or menses or catamenia or menarche).ab,kw,ti. |
|---|---|
| Search 2: Experience | (social behavior or experience or comprehension or knowledge or comprehen* or attitud* or practice* or experienc* or perception* or understand* or challenge* or barrier or facilitat* or impact or affect or effect).ab,kw,ti. |
| Search 3: Qualitative Research Method | (qualitative research or interview or qualitative* or focus group* or focus-group* or interview* or semi-structured interview* or unstructured interview* or thematic analys* or ethnograph* or grounded theory or narrative or interpretive or discourse analys* or content analys* or framework analys* or interpetiv* or interpretativ* or phenomeno* or mixed-method* or mixed method*).ab,kw,ti. |
| Search 4: Final | 1 AND 2 AND 3 |

ab = abstract, kw = keywords, ti = title.

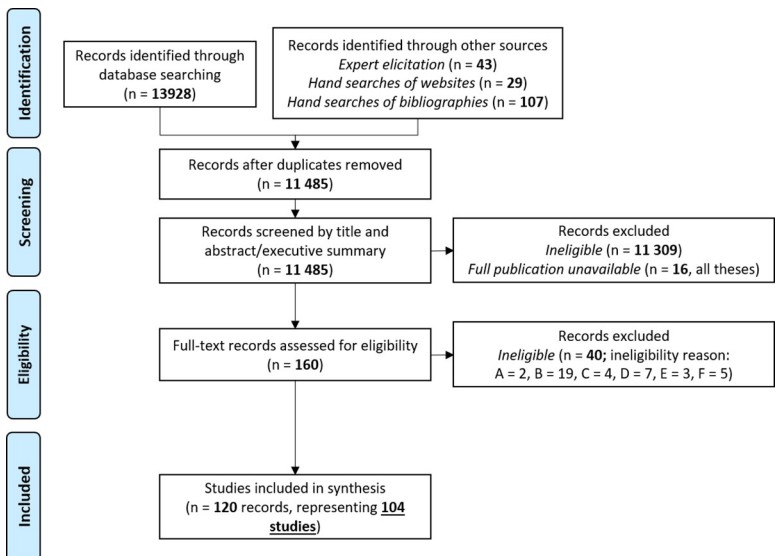

**Fig 1. Review flow diagram showing the number of titles, abstracts, and full-text records assessed for eligibility and reasons for exclusion.** Reasons for excluding full-texts were A = exclusively presented qualitative data quantitatively; B = did not report on the first person experiences of someone who menstruates; C = experience being described may have occurred in a low- or middle-income country; D = focused on non-bleeding experiences; E = did not include the full study report.

This yielded 310 websites, which were hand-searched for relevant publications and updated in November 2020 (for a full list of websites searched see S1 Text). Individual experts undertaking research focused on menstrual health (names provided in S2 Text) were contacted in October 2019 and November 2020 and asked to recommend any potentially eligible research. Reference lists of review articles and eligible publications were hand-searched (Fig 1). Every effort was made to obtain the full-texts of all potentially eligible publications, including contacting authors directly (e.g., by email, LinkedIn and ResearchGate). Full text copies were retrieved for all but 16 publications, all of which were theses.

Two authors (DJB, HJR) screened all titles and abstracts/executive summaries for eligibility. Publications were eligible for inclusion where they 1) reported on primary, qualitative research; 2) captured the personal experiences of menstruation among people residing in a HIC, as defined by the World Bank as of 2020 [17]; 3) focused on the bleeding phase of the menstrual cycle (i.e., menses/period, or withdrawal bleeding for those using hormonal contraceptives, including studies reporting on the experiences of those with endometriosis, dysmenorrhea, menorrhagia and reporting on experiences of menarche); and 4) were in English. Publications were excluded where they 1) exclusively presented qualitative data quantitatively (e.g., by back-coding qualitative data and reporting only statistics); 2) did not report on the first-person experiences of someone who menstruates (e.g., studies with only pre-menarcheal girls, expert interviews or media) or the experience being described may have occurred in a LMIC (e.g., in publications where participants were immigrants and it was unclear whether experiences described happened in LMICs or HICs); 3) focused on non-bleeding experiences, such as those primarily concerning polycystic ovary syndrome, menopause, pre-menstrual stress/tension, pregnancy, trying to conceive/fertility, birth, menstrual regulation (e.g., via hormonal contraceptives), peri-menopause/climacterium, menstrual synchrony, adolescence or sexual and reproductive health more broadly (and did not mention menstruation in the abstract/executive summary); 4) did not include the full study report (e.g., conference abstracts

without an accompanying full paper); 5) did not include description of the methodology for data collection (i.e., the reliability of the study could not be assessed).

Full text screening was undertaken by DJB. 120 publications describing 104 studies were eligible for inclusion (Fig 1).

## Study quality appraisal

Study quality was appraised using the EPPI-Centre checklist [18], developed for assessing the reliability of qualitative research based on rigor in sampling, data collection, analysis and reporting, and the usefulness of the study according to its breadth and depth, the extent to which it privileges the perspectives of those most crucial to the review, in this case, those who menstruate, and the extent to which it details results relevant to the review question. Where studies used mixed methods, we appraised the quality based only on the qualitative data collection methods, analysis and reporting. Two authors (DJB, JH) independently appraised then discussed the quality of a random 10% of studies, to calibrate quality appraisal. DJB appraised the remaining studies, with input from JH on any difficult cases. Study quality ratings and justifications are detailed in S1 Table.

## Data analysis

We used a combination of line-by-line coding and thematic network mapping to identify overarching themes and develop our final synthesis [19]. The model of menstrual experience developed through synthesis of studies in LMICs [15] served as the preliminary framework and starting point for deductive and inductive identification of themes. Aligned with the LMIC review, we sought to go beyond description and interpret findings across studies to extend our conceptual understanding of menstrual experiences [20]. Studies contributed parts to an integrated scheme, modelling menstrual experience in HICs [21].

Analysis was undertaken following 6 steps:

1. DJB familiarised herself with the included studies;

2. Using a framework approach, DJB coded the study results, quotations from those who menstruate and author interpretations, in studies of high and medium trustworthiness, in NVivo 12 [22]. Coding was deductive against the themes identified in Hennegan's review of menstrual experiences in LMICs [15], with new themes coded inductively to develop a draft coding template. Examples of a) antecedents to menstrual experiences, b) experiences related to menstrual bleeding and c) impacts on the lives of people who menstruate were coded;

3. DJB constantly compared relationships between antecedents, experiences and impact themes in high and medium trustworthiness studies and coded these relationships in NVivo 12. Relationships were considered saturated where they were evident in at least five studies, and a preliminary integrated model of menstrual experiences in HICs was developed;

4. For validation, two other authors (EW, JH) coded 15% of studies each (i.e., 30% of studies were coded in total). Using the same approach as DJB, they deductively coded studies against the framework of themes from the review of studies in LMICs and inductively identified new codes independently, that is, without having reviewed DJB's coding template. DJB reviewed co-author coding and it was consistent with the coding template;

5. The integrated model was shared with all co-authors for discussion, followed by repeated mappings until all agreed that the model developed had the greatest explanatory power;

6. DJB coded studies of low trustworthiness and assessed their fit with the final themes and integrated model. Low-quality studies supported the primary analysis, and no new constructs emerged (this is akin to the 'sensitivity analysis' common in systematic reviews of qualitative research, in which themes are checked to see if they rely on low-quality studies alone [19]).

The final model and themes are contrasted against findings from the review of LMIC studies in the Discussion, facilitated by the process of coding against these themes during our analysis.

### Positionality

The authors are women who menstruate and live in HICs. Throughout the study they employed self-critical epistemological awareness [23], considering how their own experiences influence their interpretation of findings, privileging the voices of study participants and attempting to set aside their own biases to maximise rigour in the analysis.

Prior to beginning this study, the authors had all undertaken research on menstrual health in LMIC contexts. Most of the publications included in this review were unknown to them, allowing the model to be developed from the experiences of the study participants rather than previous work the authors were familiar with.

## Results

### A note on inclusivity

Historically the terms 'girl', 'woman' and 'female' have been used interchangeably to denote individuals who menstruate. This ignores the differences between biological sex and self-identified genders. Not all people identifying as women and girls menstruate, and not everyone who menstruates identifies as a woman or girl. Although most included studies used the language of girls and women exclusively, the genders of participants may have been assumed because the participants were people who menstruate. We reviewed five studies (all published since 2016) that specifically recruited participants who menstruate but do not identify as a woman or girl, here referred to as non-binary or transgender people who menstruate [24–28]. We thus use gender neutral pronouns throughout the paper, except in cases where the finding is specifically linked to gender identity or is only relevant to cis-gendered, non-binary or transgender menstruators, in which cases we use the gendered terms and pronouns found in the original publications.

### Study characteristics

Table 2 reports study characteristics and the overall trustworthiness and relevance ratings from quality appraisal. The included studies involved over 3800 participants. Eighty-three studies included women (18 years of age and older), 24 studies included girls (below 18 years of age), and five studies included adult menstruators who identified as transgender or non-binary (18 years of age and older). Included studies spanned many decades, and frequently included retrospective reports of menarche experiences from many years prior to data collection. Table 2 provides an estimate of the time period when participants reached menarche based on their age and the date of the study. In 9 studies participants had reached menarche during the early 20th Century (C) (defined here as 1900–1949), the mid-20th C (defined here as 1950–1979) in 46 studies, the late 20th C (defined here as 1980–1999) in 61 studies and the early 21st C (defined here as after 2000) in 44 studies. We could not determine the approximate timespan of menarche for 22 studies. Twenty included studies specifically recruited

Table 2. Included studies.

| Study | Country/ Territory[1] | Population type | Number of participants[2] | Age range | Time period of menarche[3] | Decade of data collection[4] | Data collection method | Author stated analytical method and/or epistemological perspective | Trust-worthiness | Relevance |
|---|---|---|---|---|---|---|---|---|---|---|
| [5]Adams-Matthews 2009 [32] | USA | Middle-aged or older Californian women | 4 | Middle-aged or older | Pre-2000 (based on study date alone) | 2000 | Individual interviews | Interpretive phenomenological analysis | Medium | Low |
| Allen & Goldberg 2009 [33] | USA | Undergraduate students enrolled in a human sexuality course at a public university | 108 | 18–23 | Late 20th C | 2000 | Written narratives | Grounded theory | High | High |
| [6]Allyn et al. 2020 [34] | USA | Mostly female university students with dysmenorrhea | 39 | Average 21 | Early 21st C | 2010 | Individual interviews | Thematic analysis | High | High |
| Amann-Gainotti 1986 [35] | Italy | Post menarcheal girls | 85 | 11–14 | Late 20th C | 1980 | Individual interviews | Not stated | Low | Low |
| Andrews 1985 [36] | USA | Post menarcheal middle school girls | 13 | 11–13 | Late 20th C | 1980 | Individual interviews | Not stated | Medium | Medium |
| [5,8]APS Group Scotland 2018 [37] | UK | Menstruators accessing community partners of project, e.g., food banks and shelters | 28 | Unclear | After 1950 (based on study date alone) | 2010 | Interviews (individual + group) | Not stated | Medium | Medium |
| Armeni 1997 [38] | Canada | Women born between 1910–1965 | 24 | 30–85 | Early-mid 20th C | 1990 | Individual interviews | Not stated | Medium | High |
| [5,6]Armour, Dahlen & Smith 2016 [39] and Armour 2015 [40] | New Zealand | Women who participated in a randomised control trial on the effectiveness of Traditional Chinese Medicine on primary dysmenorrhea | 12 | Average 31 | Late 20th C | 2010 | Individual interviews | Thematic analysis | High | Medium |
| Artschwager 1981 [41] | USA | Mexican American women of childbearing age found at a family planning clinic | 25 | Unclear | Pre-1980 (based on study date alone) | 1980 | Individual interviews | Not stated | Low | Low |
| Beausang & Razor 2000 [42] | USA | College students enrolled in a human sexuality course at a community college in the Midwest | 85 | 18–61 | Mid-late 20th C | 1990 | Written narratives | Qualitative data analysis | Medium | Low |
| Bishop 1999 [43] | USA | Childless women recruited through community notice boards and women's clinics | 100 | 18–30 | Late 20th C | 1990 | Written questionnaire | Not stated | Medium | Low |
| Bobier 2020 [44] | USA | Private middle school students in Michigan | 9 | 9–13 | Early 21st C | 2010 | Individual interviews | Not stated | Medium | Medium |

(Continued)

**Table 2.** (Continued)

| Study | Country/ Territory[1] | Population type | Number of participants[2] | Age range | Time period of menarche[3] | Decade of data collection[4] | Data collection method | Author stated analytical method and/or epistemological perspective | Trust-worthiness | Relevance |
|---|---|---|---|---|---|---|---|---|---|---|
| Botello-Hermosa & Casado-Mejia 2015 [45] | Spain | Rural and urban women from different generations | 24 | 18+ | Unclear | 2010 | Individual interviews | Grounded theory | Medium | Medium |
| Bransen 1992 [46] | The Netherlands | White women educated to intermediate or high school level | 12 | 19–51 | Mid-late 20th C | 1990 | Interviews (individual + group) | Not stated | Low | Low |
| Brantelid, Nilver & Alehagen 2014 [47] | Sweden | Swedish women | 12 | 18–48 | Mid-late 20th C | 2010 | Individual interviews | Thematic analysis | High | Medium |
| [8]Briggs 2020 [48] | UK | Women from low-income households for whom the cost of menstruation had always been an issue and student representatives from a school council | 8 | 16+ | Unclear (ages of women not provided) | 2010 | Interviews (individual + group) | Not stated | Medium | Medium |
| Britton 1996 [49] | UK | Women who identified as white English, Irish, Trinidadian, Malaysian, Sri Lankan | 20 | 18–39 | Mid-late 20th C | 1990 | Individual interviews | Not stated | Low | Medium |
| Brookes & Tennant 1998 [50] | New Zealand | Non-indigenous New Zealand women | 50 | Unclear | Pre-2000 (based on study date alone) | 1990 | Individual interview + written narratives | Not stated | Low | Medium |
| Brown, Knight & Forrest 2020 [51] | UK | Elite female athletes in the fields of weightlifting, athletics, climbing, gymnastics and judo | 17 | 17–34 | Late 20th C + Early 21st C | 2010 | Individual interviews | Not stated | High | High |
| [6]Bullo & Hearn 2020 [52] | UK | Women with endometriosis | 21 | 23–53 | Mid-late 20th C + Early 21st C | 2010 | Individual interviews | Interpretive methodological approach + conceptual metaphor theory | High | Medium |
| [6]Burbeck & Willig 2014 [53] | UK | White women with menstrual pain | 6 | 24–36 | Late 20th C + Early 21st C | 2010 | Individual interviews | Interpretive phenomenological analysis | High | High |
| Burrows & Johnson 2005 [54] | UK | White, Western, middle-class girls | 9 | 12–15 | Early 21st C | 2000 | Group interviews | Reflexive feminist constructivist approach | Medium | Medium |
| [6]Byles, Hanrahan & Schofield 1997 [55] | Australia | Women aged 30–50 who had problems with heavy, painful or frequent periods | 16 | 33–50 | Mid 20th C | 1990 | Group interviews | Not stated | Medium | Medium |

(Continued)

**Table 2.** (Continued)

| Study | Country/ Territory[1] | Population type | Number of participants[2] | Age range | Time period of menarche[3] | Decade of data collection[4] | Data collection method | Author stated analytical method and/or epistemological perspective | Trust-worthiness | Relevance |
|---|---|---|---|---|---|---|---|---|---|---|
| Cattaneo 2000 [56] | Canada | Canadian women | 18 | Unclear | Early-late 20th C | 1990 | Individual interviews | Not stated | Medium | High |
| [6]Chapple 1999 [57] | UK | Women who report suffering menorrhagia, 13 of South Asian descent and 1 Muslim | 30 | 15–53 | Mid-late 20th C | 1990 | Individual interviews | Symbolic interactionism + social constructionism | Medium | Medium |
| [6]Chen, Draucker & Carpenter 2018 [58] | USA | Women who have experienced dysmenorrhea in past 6 months | 225 | 18–57 | Mid-late 20th C + Early 21st C | 2010 | Written questionnaire | Thematic analysis | High | Medium |
| Chou, Lu, Wang, Lan & Lin 2008 [59] | Taiwan | Institutionalized women with an intellectual disability | 55 | 21–65 | Mid-late 20th C + Early 21st C | 2000 | Individual interviews | Thematic analysis | High | Medium |
| [7]Chrisler, Gorman, Manion, Murgo, Barney, Adams-Clark, Newton & McGrath 2016 [24] | USA | Masculine of centre and transgender people | 110 | 18–64 | Mid-late 20th C + Early 21st C | 2010 | Written questionnaire | Not stated | Medium | Low |
| Christoforou 2018 [60] | Cyprus | Greek Cypriot women | 20 | 23–73 | Mid-late 20th C + Early 21st C | 2010 | Individual interviews | Grounded theory | Medium | High |
| [6]Clark 2012 [61] | UK | Members of an Endometriosis UK support group | 13 | 23–46 | Mid-late 20th C + Early 21st C | 2010 | Individual interviews | Interpretive phenomenological analysis | High | Medium |
| [8]Cooper & Koch 2007 [62] | USA | African-American women from a public housing project | 17 | 18–50 | Mid-late 20th C + Early 21st C | 2000 | Interviews (individual + group) | Grounded theory | Low | Low |
| Costos, Ackerman, & Paradis 2002 [63] | USA | Women with a Bachelor's degree level of education | 138 | 26–60 | Mid-late 20th C | 1990 | Individual interviews | Content analysis | Medium | Medium |
| Deforest 2007 [64] | USA | Women | 7 | 20–40 | Mid-late 20th C + Early 21st C | 2000 | Individual interviews via email | Interpretive phenomenological analysis | Low | Medium |
| [8]DeMaria, Delay, Sundstrom, Rehberg, Naoum, Ramos-Ortiz, Meier & Brig 2019 [65] | USA | Women living in South Carolina who speak English or Spanish | 70 | 19–78 | Mid-late 20th C + Early 21st C | 2010 | Individual interviews | Grounded theory | High | Medium |

*(Continued)*

**Table 2.** (Continued)

| Study | Country/ Territory[1] | Population type | Number of participants[2] | Age range | Time period of menarche[3] | Decade of data collection[4] | Data collection method | Author stated analytical method and/or epistemological perspective | Trust-worthiness | Relevance |
|---|---|---|---|---|---|---|---|---|---|---|
| DeMaria, Meier & Dykstra 2019 [66] | Italy | Reproductive age women living in or around Florence, Italy, utilizing the Italian healthcare system, and fluent in conversational English. | 46 | 18–45 | Late 20th C + Early 21st C | 2010 | Individual interviews | Grounded theory | Medium | Low |
| Denny, Culley, Papadopoulos & Apenteng 2011 [67] | UK | Healthy women from 5 minority ethnic groups | 42 | 18+ | Unclear | 2000 | Group interviews | Framework analysis | High | High |
| Dillaway, Cross, Lysack & Schwartz 2013 [68] | USA | Women living with spinal cord injuries | 20 | 27–66 | Mid-late 20th C | 2010 | Individual interviews | Thematic analysis | High | Medium |
| Ditchfield & Burns 2004 [69] | UK | Learning-disabled women | 11 | 20–44 | Late 20th C | 2000 | Individual interviews | Thematic analysis | Medium | Medium |
| Donmall 2013 [70] | UK | White, British women | 6 | 21–25 | Early 21st C | 2010 | Individual interviews | Discursive psychology + thematic analysis + narrative analysis | Medium | Medium |
| [6]Elson 2002 [71] | USA | Women who have undergone premenopausal hysterectomy | 40 | 24–97 | Early-late 20th C | 2000 | Individual interviews | Grounded theory | Medium | Medium |
| Eriksen 2016 [72] | USA | Women with Autism spectrum disorder | 10 | 13–16 | Early 21st C | 2010 | Interviews (individual + group) | Thematic analysis | Medium | Medium |
| Fahs 2011 [73] & Fahs 2014 [74] & Fahs 2020 [75] | USA | Women, sexual minority women and racial/ethnic minority women were intentionally oversampled | 40 | 19–54 | Mid-late 20th C | 2000 + 2010 | Individual interviews | Thematic analysis | High | High |
| [6]Fernández-Martínez, Abreu-Sánchez, Pérez-Corrales, Ruiz-Castillo, Velarde-García & Palacios-Ceña 2020 [76] | Spain | Nursing students with primary dysmenorrhea | 33 | Average 23 | Early 21st C | 2010 | Group interviews | Thematic analysis | High | High |
| Findlay, Macrae, Whyte, Easton & Forrest 2020 [77] | UK | International female rugby players | 15 | Average 25 | Early 21st C | 2010 | Individual interviews | Thematic analysis | High | High |

(Continued)

**Table 2.** (Continued)

| Study | Country/ Territory[1] | Population type | Number of participants[2] | Age range | Time period of menarche[3] | Decade of data collection[4] | Data collection method | Author stated analytical method and/or epistemological perspective | Trust-worthiness | Relevance |
|---|---|---|---|---|---|---|---|---|---|---|
| Fingerson 2005 [78] & Fingerson 2006 [79] | USA | Mostly white, high school age girls in the US | 26 | 13–18 | Late 20th C + Early 21st C | 1990 + 2000 | Interviews (individual + group) | Grounded theory | Medium | Medium |
| Fitzgerald 2015 [80] | Ireland | Women living in the Republic of Ireland | 19 | 22–58 | Mid-late 20th C + Early 21st C | 2010 | Individual interviews | Grounded theory | Medium | High |
| [7]Frank 2020 [25] | USA | Trans and non-binary menstruators | 19 | 18–29 | Late 20th C + Early 21st C | 2010 | Individual interviews | Ethnographic content analysis | Medium | Medium |
| Freidenfelds 2009 [81] | USA | Primarily African-Americans in rural South, white Americans in New England, Chinese Americans in California | 75 | 18–90+ | Early-late 20th C + Early 21st C | 1990 | Individual interviews | Not stated | Medium | High |
| George & Murcott 1992 [82] | UK | Women students attending a business studies course at a tertiary college in South Wales | 20 | 16–18 | Late 20th C | 1980 | Individual interviews | Not stated | Low | Medium |
| Golub, & Catalano 1983 [83] | USA | Women college students and women aged 30–45 | 137 | 18–45 | Early-late 20th C | 1970 | Written questionnaire | Not stated | Low | Low |
| Goolden 2018 [84] | UK | Women who work in menstrual health and hygiene specifically or sexual and reproductive health more broadly | 11 | Unclear | Unclear | 2010 | Individual interviews | Interpretive phenomenological analysis | Medium | Low |
| [6]Grundström, Alehagen, Kjølhede & Berterö 2018 [85] | Sweden | Women with a laparoscopy-confirmed diagnosis of endometriosis | 9 | 23–55 | Mid-late 20th C + Early 21st C | 2010 | Individual interviews | Interpretive phenomenological analysis | Medium | Medium |
| Hawkey, Ussher, Perz & Metusela 2017 [86] | Australia and Canada | Women from Afghanistan, Iraq, Somalia, South Sudan, Sudan, Sri Lanka, and varying South American countries, who have settled in Sydney or Vancouver | 82 | 18–70 | Mid-late 20th C + Early 21st C | 2010 | Interviews (individual + group) | Thematic decomposition | High | Medium |

(*Continued*)

**Table 2.** (Continued)

| Study | Country/ Territory[1] | Population type | Number of participants[2] | Age range | Time period of menarche[3] | Decade of data collection[4] | Data collection method | Author stated analytical method and/or epistemological perspective | Trust-worthiness | Relevance |
|---|---|---|---|---|---|---|---|---|---|---|
| Jackson 2019 [87] | USA | A group of urban, working-class women | 13 | 11–16 | Early 21st C | 2010 | Interviews (individual + group) | Thematic analysis | Medium | Medium |
| Jackson & Falmagne 2013 [88] | USA | University students | 13 | 18–21 | Early 21st C | 2010 | Individual interviews | Feminist interpretive | Medium | Medium |
| Kalman 2003a [89] & Kalman 2003b [90] | USA | English-speaking, adolescent girls self-identified as living with their fathers as primary caregiver. | 10 | 10–18 (including pre-menarcheal participants) | Late 20th C + Early 21st C | 1990 | Individual interviews | Grounded theory | Medium | Medium |
| Kissling 1996 [91] | USA | Girls recruited through scout troops and school guidance counsellors | 8 | 12–16 | Late 20th C | 1990 | Interviews (individual, group, mother-daughter) | Critical feminist analysis | High | Medium |
| Koutroulis 2001 [92] | Australia | Friends, or relatives of friends, of the author | 8 | 30–50 | Mid-late 20th C | 1990 | Individual interviews | Cross-sectional approach to memory work | Low | Low |
| Lee 1994 [93] & Lee & Sasser-Coen 1996 [94] Sasser-Coen 1997 [95] | USA | Volunteers recruited via public advertising | 40 | 18–94 | Early-late 20th C | 1990 | Individual interview + written narratives | Interpretive phenomenological analysis | High | Medium |
| Lee 2008 [96] & Lee 2009 [97] | USA | Students enrolled in introductory gender classes | 155 | 18–21 | Late 20th C + Early 21st C | 2000 | Written narrative | Narrative analysis | High | Medium |
| Lee 2002 [98] | Canada | Women | 43 | 19–55 | Mid-late 20th C | 1990 | Individual interviews | Narrative analysis | Medium | Medium |
| [6]Li, Bellis, Girling, Jayasinghe, Grover, Marino & Peate 2020 [99] | Australia | Adolescent girls presenting with heavy menstrual bleeding and/or dysmenorrhea at the clinic | 30 | 12–18 | Early 21st C | 2010 | Individual interviews | Grounded theory | High | Medium |
| [7]Lowik 2020 [26] | Canada | Trans and non-binary individuals | 10 | 20s-60s | Mid-late 20th C + Early 21st C | 2010 | Individual interviews + PhotoVoice | Thematic narrative analysis | High | Medium |
| [6]Marshall 1998 [100] | UK | Selected by a gynaecologist on the basis of referral letters from GPs which indicated that these women were experiencing heavy menstrual bleeding | 23 | 21–53 | Mid-late 20th C | 1990 | Individual interviews | Thematic analysis | Medium | Medium |

(Continued)

**Table 2.** (Continued)

| Study | Country/ Territory[1] | Population type | Number of participants[2] | Age range | Time period of menarche[3] | Decade of data collection[4] | Data collection method | Author stated analytical method and/or epistemological perspective | Trust- worthiness | Relevance |
|---|---|---|---|---|---|---|---|---|---|---|
| Marshall, Dasari, Nathaniel, Grill, Nichols & Pruthi 2019 [101] | USA | Women diagnosed with Von Willebrand's Disease previously seen in the clinic | 82 | 33–64 | Mid-late 20th C | 2010 | Individual interviews | Thematic analysis | Low | Low |
| Mason & Cunningham 2008 [102] | UK | Women with Down syndrome | 6 | 14–40 (but this might include age of Down syndrome daughters where mother was interviewed) | Unclear | 2000 | Individual interviews | Thematic analysis | Medium | Low |
| [6]Matías-González, Sánchez-Galarza, Flores-Caldera & Rivera-Segarra 2020 [103] | Puerto Rico | Puerto Rican women with a diagnosis of endometriosis | 50 individuals in 5 focus groups | 21+ | Unclear | 2010 | Group interviews | Thematic analysis | Medium | Medium |
| McKechnie 2000 [104] | UK | Women who had consulted GP about menstrual irregularity | 29 | Unclear | Pre-2000 (based on study date alone) | 1990 | Individual interviews | Not stated | Low | Low |
| Moas 2010 [105] | Israel | Fertile Jewish-Israeli women who are "menstrually aware" | 19 | 26–46 | Mid-late 20th C | 2000 | Individual interviews | Thematic analysis | High | Medium |
| Murray 1996 [106], Murray 1997 [107], Murray 1998 [108] | Australia | Australian women from a variety of social and ethnic backgrounds | 20 | 50s-80s | Early-mid 20th C | 1990 | Individual interviews | Not stated | Medium | Medium |
| Newton 2012 [109] & Newton 2016 [110] | UK | Sample of women and girls in North Midlands (English) town | Unclear | Large range from children to elderly | Unclear | 2000 | Interviews (individual + group) + written questionnaire | Thematic analysis | Medium | Medium |
| [6]O'Flynn & Britten 2000 [111] & O'Flynn 2006 [112] | UK | Women who had consulted GPs about heavy periods + women with and without menstrual complaints living in inner-city London | 21 | 18–57 | Mid-late 20th C | 1990 + 2000 | Individual interviews | Framework analysis | High | High |
| Oinas 1999 [113] | Finland | Women students of Women's Studies at a Finnish university | 8 | Unclear | Mid-late 20th C | 1990 | Written narrative + group interviews | Memory work method | Low | Low |

(*Continued*)

**Table 2.** (Continued)

| Study | Country/ Territory[1] | Population type | Number of participants[2] | Age range | Time period of menarche[3] | Decade of data collection[4] | Data collection method | Author stated analytical method and/or epistemological perspective | Trust-worthiness | Relevance |
|---|---|---|---|---|---|---|---|---|---|---|
| Owen 2020 [114] | Australia | Undergraduate or recent graduate women | 11 | 20–24 | Early 21st C | 2010 | Individual interviews + diaries | Feminist ethnography | High | High |
| Owen 2020 [114] | UK | Women employees or board members of small company developing/ implementing a menstrual leave policy | 12 | 27–43 (age range includes one man) | Unclear | 2010 | Interviews (individual + group) | Feminist ethnography | High | Medium |
| Pafford 2007 [115] | USA | Women who have military service and experience in austere field environments | 7 women, direct observations were made of many individuals | 43–55 | Mid 20th C | 2000 | Individual interviews + direct observation | Ethnography | Medium | Low |
| Pascoe 2007 [116] & Pascoe 2015 [117] | Australia | Australian women across generations, known to author | 13 | 20s – 90s | Early-late 20th C | 2000 | Individual interviews | Not stated | Medium | Medium |
| Patterson & Hale 1985 [118] | USA | Volunteers of a women's association | 25 | Unclear | Pre 2000 (based on study date alone) | 1980 | Individual interviews, direct observations + anecdotes | Not stated | Medium | Medium |
| [5,6]Prileszky 2013 [119] | UK | Women consulting health professionals about heavy menstrual bleeding | 27 | 25–51 | Mid-late 20th C | 2000 | Individual interviews | Grounded theory | High | Medium |
| [7]Raynor 2020 [27] | USA | White individuals in central Texas who identify as transgender or gender queer | 3 | 22–42 | Late 20th C + Early 21st C | 2010 | Individual interviews | Interpretive phenomenological analysis | Medium | High |
| Rodgers 2001 [120] | UK | Women with learning difficulties | 21 | Unclear | Unclear | 1990 | Individual interviews | Grounded theory | High | Medium |
| Rubinsky, Gunning & Cooke-Jackson 2020 [121] | USA | Women | 165 | 18–37 | Late 20th C + Early 21st C | 2010 | Written questionnaire | Not stated | High | High |
| [7]Rydström 2018 [28] | Sweden | Trans and non-binary menstruators | 9 | 19–32 | Late 20th C + Early 21st C | 2010 | Individual interviews | Inductive-abductive analysis | High | High |
| [6]Santer 2005 [122], Santer, Wyke & Warner 2008 [123] & Santer, Wyke & Warner 2008 [124] | UK | Women who responded to a postal survey saying that they experience heavy menstrual bleeding | 32 | 27–45 | Mid-late 20th C | 2000 | Individual interviews | Thematic analysis | High | High |

(*Continued*)

**Table 2.** (Continued)

| Study | Country/Territory[1] | Population type | Number of participants[2] | Age range | Time period of menarche[3] | Decade of data collection[4] | Data collection method | Author stated analytical method and/or epistemological perspective | Trust-worthiness | Relevance |
|---|---|---|---|---|---|---|---|---|---|---|
| [6]Scott, Hintz & Harris 2020 [125] | USA | Women with chronic pelvic and genital pain conditions | 17 | 18–30 | Late 20th C + Early 21st C | 2010 | Individual interviews | Critical feminist theorising | Medium | Low |
| [8]Sebert Kuhlmann, Peters Bergquist, Danjoint & Wall 2019 [126] | USA | Women recruited through community organisations offering services for low income individuals | 183 interviews, 17 in focus groups | Average 35.8 | Unclear | 2010 | Interviews (individual + group) | Descriptive analysis | High | Medium |
| Secor-Turner, Huseth-Zosel & Ostlund 2020 [127] | USA | Adolescent girls in middle and high school who had already experienced their first menstrual period | 12 | 12–16 | Early 21st C | 2010 | Group interviews | Descriptive content analysis | Medium | Medium |
| [6]Seear 2009 [128] | Australia | Women with endometriosis | 20 | 24–55 | Mid-late 20th C | 2000 | Individual interviews | Interactive model | Medium | Medium |
| [6]Segal [129] | USA | Women with endometriosis | 5 | Unclear | Pre-2000 (based on study date alone) | 1990 | Individual interview + written narratives | Interpretive phenomenological analysis | Low | Low |
| Skultans 1970 [130] & Skultans 1988 [131] | UK | Women living in a small Welsh village | 36 | 29–71 | Early-mid 20th C | 1970 | Individual interviews | Not stated | Low | Low |
| [8]Sommer, Gruer, Clark Smith, Morko & Hopper 2020 [132] | USA | People who menstruate experiencing homelessness (including living on the streets and in shelters) in New York City | 22 | 18–62 (approximated by authors) | Mid-late 20th + Early 21st C | 2010 | Individual interviews | Thematic analysis | High | Medium |
| Statham 2020 [133] | UK | Elite athletes, Olympic level athletes | 4 | Unclear | Unclear | 2010 | Individual interviews | Thematic analysis | Medium | Low |
| Steward, Crane, Mairi Roy, Remington & Pellicano 2018 [134] | Online/global | Autistic and non-autistic women | 237 | 16–60 | Mid-late 20th C + Early 21st C | 2010 | Written questionnaire | Thematic analysis | Medium | Low |
| Teitelman 2004 [135] | USA | African American, European American, and multiethnic girls | 22 | 14–18 | Late 20th C + Early 21st C | 1990 | Individual interviews | Interpretive methodological approach | High | Medium |
| Thuren 1994 [136] | Spain | Valencian women known to the author | 36 | Unclear | Pre-2000 (based on study date alone) | 1990 | Individual interviews | Not stated | Low | Low |

(Continued)

**Table 2.** (Continued)

| Study | Country/Territory[1] | Population type | Number of participants[2] | Age range | Time period of menarche[3] | Decade of data collection[4] | Data collection method | Author stated analytical method and/or epistemological perspective | Trust-worthiness | Relevance |
|---|---|---|---|---|---|---|---|---|---|---|
| Tingle & Vora [8] | UK | Young women in England and Northern Ireland (NI) | 56 girls in England, unclear how many were girls in NI | 9–24 | Early 21st C | 2010 | Group interviews | Not stated | Low | High |
| Tolson, Fleming & Schartau 2002 [137] | UK | Women with Parkinson's disease | 19 | 34–56 | Mid-late 20th C | 1990 | Interviews (individual and group) + diaries + creative writing | Not stated | High | Medium |
| Trego 2007 [138] | USA | Women in the US Army who had been deployed to U.S. military operations in South and Central Asia, Iraq, the Arabian Peninsula, and the Horn of Africa | 9 | Unclear | Pre-2000 (based on study date alone) | 2000 | Individual interviews | Content analysis | Medium | Medium |
| Uskul 2004 [139] | Global | Women from a variety of high income countries attending a summer school | 17 | 23–52 (but this range includes those from LMICs) | Unclear | 2000 | Group interviews | Thematic analysis | High | Medium |
| [8]Vora 2017 [9] & Vora 2020 [10] | UK | Women accessing a range of services that support vulnerable people in precarious housing situations | 40 | Unclear | Unclear | 2010 | Individual interviews | Not stated | Low | Low |
| Whisnant & Zegans 1975 [140] | USA | Campers and counsellors at a non-denominational overnight camp | 15 post menarcheal campers, 10 counsellors | 12–21 | Mid 20th C | 1970 | Individual interviews | Not stated | Low | Medium |
| Wigmore-Sykes, Ferris and Singh 2020 [141] | UK | Students from a school in Warwickshire | 11 | 16–18 | Early 21st C | 2010 | Interviews (individual and group) | Thematic analysis | Low | Low |
| Wood, Koch & Mansfield 2007 [142] | USA | Students from an upper level human sexuality college class | 15 | 18–22 | Late 20th C + Early 21st C | 2000 | Individual interviews | Feminist grounded theory | Medium | Medium |

(Continued)

**Table 2.** (Continued)

| Study | Country/ Territory[1] | Population type | Number of participants[2] | Age range | Time period of menarche[3] | Decade of data collection[4] | Data collection method | Author stated analytical method and/or epistemological perspective | Trust-worthiness | Relevance |
|---|---|---|---|---|---|---|---|---|---|---|
| Wootton & Morison 2020 [143] | New Zealand | Economically deprived young women in Aotearoa, recruited from two low-income high schools. All participants identified as Māori or part- Māori | 12 | 16–17 | Early 21st C | 2010 | Individual interviews | Narrative discursive approach | Medium | Medium |

[1] Taiwan and Puerto Rico are classified as separate economies from China and the US, respectively, by the World Bank, and are thus separated here.

[2] Indicates number of participants (i.e., not number of group interviews, etc.). *n*'s reported reflect only the population of interest (those who menstruate where the menstrual experience has taken place in a high income country) and do not include data collected from caregivers, boys/men or pre-menarcheal girls.

[3] Time periods of menarche included early 20th Century (C) (defined here as early 20th Century (C) (defined here as 1900–1949), mid-20th C (defined here as 1950–1979), late 20th C (defined here as 1980–1999) and early 21st C (defined here as after 2000).

[4] Decade of data collection is an estimate where publications were unclear.

[5] Indicates an intervention study.

[6] Indicates a study where participants with menstrual disorders were specifically recruited.

[7] Indicates a study where those who menstruate but identify as transgender or non-binary were specifically recruited.

[8] Indicates a study where low-income participants were specifically recruited.

participants experiencing endometriosis, dysmenorrhea and/or menorrhagia. Six studies specifically recruited low-income status participants.

North America (47 studies) and Europe (42 studies) were represented far more than other regions (Oceania = 11 studies, Asia = 1 study, Middle East = 1 study, global/online = 2 studies). Eighty-six studies collected data using individual interviews (85 verbally, 1 written), 23 used group interviews (including focus group discussions), seven written narratives, seven written questionnaires, two direct observations, two written diaries and one creative writing. Four studies involved interventions which aimed to improve experiences of menstruation. Included studies were situated within the disciplines of population health, sociology or gender studies. Most approached data collection and analysis through a social constructivist lens, and sought to understand menstrual experience through an extensively detailed understanding of each participant's lived experience through the use of interpretive phenomenological analysis [29], grounded theory [30] or thematic analysis [31].

## Study quality

Study quality was varied, with 36 studies rated as high, 48 as medium, and 20 as low trustworthiness (detailed in S1 Table). Lower-quality studies were characterised by small, convenience samples and limited details on data collection and analysis.

Twenty studies were rated as highly relevant, 59 as medium, and 25 as low. Most studies rated as high relevance had findings which were reflective of a large proportion of the population (e.g., menstruators in HICs), whereas studies rated as medium relevance were more likely to be specific to the experiences of sub-populations (e.g., menstruators with an intellectual disability) or during specific experiences (e.g., at menarche). Studies rated as low relevance did not clearly incorporate or represent the voices of those who menstruate in study design and/or findings.

## Developing an integrated model of menstrual experience

Fig 2 presents the final integrated model of menstrual experience in HICs, summarising the major themes and the relationships between them. Table 3 details which studies contributed to each theme. Boxes 1–3 provide quotations which support the findings for each theme.

## Antecedents of menstrual experience

**Socio-cultural context.** The majority of studies highlighted the importance of the socio-cultural context in shaping menstrual experiences [8, 9, 24, 26, 28, 32, 33, 36, 38–40, 42, 43, 46–49, 51, 53–61, 63–65, 67–73, 78–81, 84–90, 92–98, 101, 103, 105, 106, 108–112, 114, 116, 117, 119–122, 127–130, 132, 135–137, 139, 140]. Specifically, more than half of studies described menstruation as a stigmatised topic [8, 9, 24, 28, 32, 33, 38, 40, 42, 47–49, 53–56, 58–61, 63, 64, 67–73, 78–81, 84–90, 92–98, 101, 105, 108–112, 114, 116, 117, 119–121, 127–130, 132, 135–137, 139], with authors finding the menstruating body to be considered "a body that is polluting and potentially dangerous for women themselves, for others, and for anything sacred" [pg. 49, 60]. The construction of menstruation as polluting or dirty conflicted with gendered expectations that women and girls should be clean and feminine. Menstruation was thus embarrassing and required concealment. The socio-cultural context of menstrual stigma and gender norms across all studies manifested in strong behavioural expectations for menstruation.

Menarche was often viewed as part of 'becoming a woman' [8, 24, 38, 41, 42, 46, 47, 49, 50, 53–57, 59–61, 63, 64, 67–72, 78–82, 86–97, 106, 108–110, 112, 116–120, 122, 124, 128, 129, 135, 137, 138, 140–142]. For most participants this invoked negative emotional responses, but

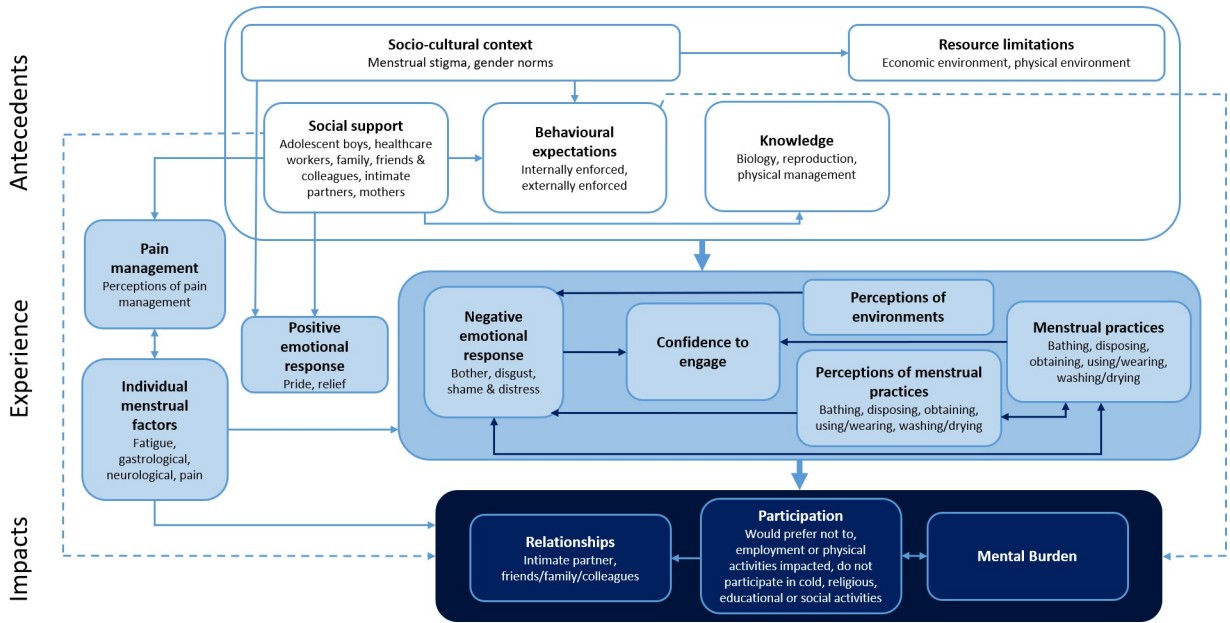

**Fig 2. Integrated model of menstrual experience.** Bolded text represents major themes, unbolded text describes sub-themes. Arrows depict directional and bidirectional relationships between themes.

some reported positive emotions associated with growing up. For menstruators who identify as non-binary or transgender, menstruation's signification of womanhood often triggered gender dysphoria, as they struggled to reconcile this construction of menstruation with their personal experiences of their bodies [24, 25, 27, 28].

**Behavioural expectations.**   Included studies described a range of expectations that influenced how participants experienced and behaved during menstruation, and the impact on their lives. These expectations could be externally enforced, where someone other than the participant was insisting that they do or say something [8, 32, 33, 35, 36, 38, 42, 45, 49, 51, 54, 56, 58, 60, 63–67, 71, 73, 78, 80, 81, 84, 86, 89, 92–97, 105, 106, 108–110, 112, 116, 117, 119–121, 125, 128, 136, 143]; or they could be internally enforced, where adherence was driven by participants' personal beliefs and internalised expectations [8, 28, 32, 33, 38, 40, 41, 47–51, 53–56, 58–60, 64–68, 71, 73–81, 84, 86–89, 93–97, 105–112, 116–122, 125, 128, 129, 133, 139, 143].

Many studies outlined expectations to keep menstruation secret through hiding menstrual materials, concealing odour, buying products discreetly or not speaking about menstruation [8, 38, 42, 51, 54, 56, 60, 63, 64, 66, 80, 81, 86, 89, 94, 95, 97, 108, 110, 112, 117, 120, 121, 125, 128]. This was commonly enforced through instructions from mothers on how to hide menstruation and to not speak about it [38, 42, 60, 63, 86, 94, 95, 115–117, 121, 125]. Mothers also placed other restrictions on the participation of daughters in a variety of activities once they had begun menstruating, further discussed in the section describing impacts on participation.

Study participants often stated that they hid evidence of menstruation, including the physical symptoms and pain management associated with it, because they believed this was socially expected of them, even where they had not been specifically told to do so [8, 32, 38, 41, 47, 50, 53–56, 58, 60, 64, 67, 75–81, 87, 88, 93–96, 106, 108–110, 112, 116–122, 128, 129, 143]. Where their menstruation did become obvious to others this resulted in strong negative emotions, feeling distressed and embarrassed. Participants generally did not speak about menstruation (their own or others') publicly, particularly not with men and boys, as they considered this to

**Table 3. Summary table of studies contributing to each theme according to trustworthiness.**

| Theme | High trustworthiness (n = 36 studies) | Medium trustworthiness (n = 48 studies) | Low trustworthiness (n = 20 studies) |
|---|---|---|---|
| **ANTECEDENTS** | | | |
| **Socio-cultural context (n = 71 studies)** | n = 29 studies [26, 28, 33, 39, 40, 47, 51, 53, 58, 59, 61, 65, 67, 68, 73, 86, 93–97, 105, 111, 112, 114, 119–122, 132, 135, 137, 139] | n = 31 studies [24, 32, 36, 38, 42, 43, 48, 54–57, 60, 63, 69–72, 78–81, 84, 85, 87–90, 98, 103, 106, 108–110, 116, 117, 127, 128] | n = 11 studies [8, 9, 46, 49, 64, 92, 101, 129, 130, 136, 140] |
| | *Menstrual stigma*, n = 24 studies [28, 33, 40, 47, 53, 58, 59, 61, 67, 68, 73, 86, 93–97, 105, 111, 112, 114, 119–121, 132, 135, 137, 139] | *Menstrual stigma*, n = 28 studies [24, 32, 38, 42, 48, 54–56, 60, 63, 69–72, 78–81, 84, 85, 87–90, 98, 108–110, 116, 117, 127, 128] | *Menstrual stigma*, n = 9 studies [8, 9, 49, 64, 92, 101, 129, 130, 136] |
| | *Gender norms*, n = 23 studies [26, 28, 33, 39, 40, 47, 51, 58, 59, 61, 65, 67, 73, 93–97, 105, 111, 114, 119–122, 132, 135] | *Gender norms*, n = 26 studies [24, 32, 36, 38, 43, 54–57, 60, 63, 69, 71, 78, 80, 81, 85, 87, 88, 98, 103, 106, 108–110, 116, 117, 127, 128] | *Gender norms*, n = 7 studies [46, 49, 64, 92, 101, 129, 140] |
| **Behavioural expectations (n = 69 studies)** | n = 24 studies [28, 33, 40, 47, 51, 53, 58, 59, 65, 67, 68, 73–77, 86, 93–97, 105, 111, 112, 119–122, 139] | n = 26 studies [32, 36, 38, 42, 45, 48, 54–56, 60, 63, 66, 71, 78, 79, 81, 84, 87–89, 106–110, 116–118, 125, 128, 133, 143] | n = 9 studies [8, 35, 41, 49, 50, 64, 92, 129, 136] |
| | *Externally enforced*, n = 15 studies [33, 51, 58, 65, 67, 73, 86, 93–97, 105, 112, 119–121] | *Externally enforced*, n = 22 studies [32, 36, 38, 42, 45, 54, 56, 60, 63, 66, 71, 78, 80, 81, 84, 89, 106, 108–110, 116, 117, 125, 128, 143] | *Externally enforced*, n = 6 studies [8, 35, 49, 64, 92, 136] |
| | *Internally enforced*, n = 24 studies [28, 33, 40, 47, 51, 53, 58, 59, 65, 67, 68, 73–77, 86, 93, 95–97, 105, 111, 112, 119–122, 139] | *Internally enforced*, n = 22 studies [32, 38, 48, 54–56, 60, 65–67, 71, 78–81, 84, 87–89, 93, 94, 106–110, 116–118, 125, 128, 133, 143] | *Internally enforced*, n = 6 studies [8, 41, 49, 50, 64, 129] |
| **Social support (n = 83 studies)** | n = 27 studies [26, 28, 33, 34, 40, 47, 51, 58, 59, 61, 65, 67, 73, 75–77, 93–97, 99, 105, 111, 112, 114, 119–122, 124, 135, 137] | n = 39 studies [24, 25, 27, 32, 36–38, 42, 43, 48, 54–57, 60, 63, 66, 69–72, 78–81, 84, 85, 87–90, 100, 102, 103, 106, 108–110, 115–118, 125, 127, 128, 138, 143] | n = 17 studies [8–10, 35, 46, 49, 50, 62, 64, 82, 83, 101, 104, 113, 129, 136, 140, 141] |
| | *Adolescent boys*, n = 7 studies [42, 75, 93, 95–97, 99, 114, 121] | *Adolescent boys*, n = 10 studies [32, 54, 78, 81, 88–90, 110, 117, 127, 143] | *Adolescent boys*, n = 2 studies [35, 141] |
| | *Healthcare workers*, n = 16 studies [26, 28, 40, 51, 58, 59, 61, 67, 75, 77, 99, 111, 112, 119, 120, 122, 137] | *Healthcare workers*, n = 13 studies [27, 55–57, 63, 71, 80, 81, 85, 100, 103, 106, 128] | *Healthcare workers*, n = 5 studies [8, 46, 64, 101, 129] |
| | *Family (not including mothers)*, n = 11 studies [51, 61, 65, 67, 95, 99, 111, 119, 121, 122, 135] | *Family (not including mothers)*, n = 17 studies [25, 32, 38, 43, 54, 56, 57, 60, 63, 71, 87, 89, 90, 103, 106, 109, 110, 127] | *Family (not including mothers)*, n = 6 studies [35, 62, 113, 136, 140, 144] |
| | *Friends and colleagues*, n = 19 studies [28, 40, 47, 51, 61, 65, 67, 75, 76, 94, 95, 97, 99, 105, 112, 114, 119, 121, 122, 124, 135] | *Friends and colleagues*, n = 23 studies [24, 25, 32, 37, 38, 43, 48, 56, 63, 71, 78, 79, 81, 87, 88, 103, 106, 110, 115, 118, 127, 128, 138] | *Friends and colleagues*, n = 7 studies [8–10, 35, 64, 129, 136, 140] |
| | *Intimate partners*, n = 10 studies [33, 34, 51, 61, 73, 75, 112, 114, 119, 122, 137] | *Intimate partners*, n = 8 studies [25, 32, 36, 38, 42, 43, 54, 56, 57, 63, 69–72, 79, 81, 84, 88, 102, 106–110, 116, 125, 127, 128] | *Intimate partners*, n = 1 study [64] |
| **Knowledge (n = 57 studies)** | n = 20 studies [33, 39, 47, 51, 58, 59, 67, 74, 75, 86, 91, 93–97, 99, 105, 114, 119–122, 135] | *Mothers*, n = 25 studies [25, 32, 36, 38, 42, 43, 54, 56, 57, 63, 69–72, 79, 81, 84, 88, 102, 106–110, 116, 125, 127, 128] | *Mothers*, n = 13 studies [8, 35, 49, 50, 62, 64, 82, 83, 104, 113, 129, 140, 141] |
| | *Mothers*, n = 14 studies [34, 47, 61, 65, 76, 93–96, 99, 114, 119–122, 135] | n = 27 studies [32, 36, 38, 42, 43, 45, 55–57, 60, 63, 69–72, 78, 81, 85, 87–89, 102, 106, 109, 110, 116, 117, 127, 128] | n = 10 studies [8, 35, 41, 50, 64, 104, 130, 136, 140, 141] |
| | *Biology*, n = 16 studies [39, 58, 59, 65, 67, 74, 75, 91, 93–97, 105, 114, 119–122, 135] | *Biology*, n = 24 studies [32, 36, 38, 42, 45, 55–57, 60, 63, 69–72, 81, 85, 87, 89, 102, 106, 109, 110, 116, 127, 128] | *Biology*, n = 10 studies [8, 35, 41, 50, 64, 104, 130, 136, 141] |
| | *Reproduction*, n = 5 studies [33, 58, 74, 95, 120] | *Reproduction*, n = 8 studies [32, 56, 60, 63, 71, 78, 81, 106] | *Reproduction*, n = 1 study [41] |
| | *Physical management*, n = 16 studies [39, 47, 51, 58, 86, 91, 94–96, 99, 105, 114, 119–122, 135] | *Physical management*, n = 17 studies [32, 36, 38, 42, 43, 56, 63, 69, 71, 72, 81, 88, 106, 110, 116, 117, 127] | *Physical management*, n = 4 studies [8, 50, 104, 140] |

*(Continued)*

Table 3. (Continued)

| Theme | High trustworthiness (n = 36 studies) | Medium trustworthiness (n = 48 studies) | Low trustworthiness (n = 20 studies) |
|---|---|---|---|
| **Resource limitations (n = 31 studies)** | n = 13 studies [26, 28, 40, 47, 51, 94, 111, 119, 120, 122, 126, 132] | n = 16 studies [24, 25, 37, 48, 54, 56, 78, 80, 81, 88, 98, 106, 108, 115, 117, 138, 143] | n = 2 studies [8–10] |
| | *Economic environment*, n = 5 studies [40, 114, 119, 120, 126] | *Economic environment*, n = 8 studies [37, 48, 56, 81, 98, 106, 117, 143] | *Economic environment*, n = 2 studies [8–10] |
| | *Physical environment*, n = 10 studies [26, 28, 47, 51, 94, 111, 120, 122, 126, 132] | *Physical environment*, n = 10 studies [25, 54, 78, 80, 81, 88, 106, 108, 115, 117, 138] | *Physical environment*, n = 1 study [10] |
| **EXPERIENCES** | | | |
| **Menstrual practices (n = 71 studies)** | n = 24 studies [28, 33, 47, 51, 59, 67, 68, 75, 76, 86, 91, 93–96, 99, 105, 112, 114, 119–123, 126, 132, 135, 137] | n = 34 studies [24, 25, 32, 36–38, 42, 44, 48, 54, 56, 57, 60, 63, 66, 69, 71, 72, 78–81, 84, 88, 89, 102, 106, 108–110, 115–118, 128, 138, 142, 143] | n = 13 studies [8–10, 41, 46, 50, 64, 82, 83, 92, 104, 113, 136, 140] |
| | *Bathing*, n = 7 studies [47, 68, 86, 95, 105, 120, 132] | *Bathing*, n = 7 studies [32, 38, 56, 60, 66, 138, 143] | *Bathing*, n = 2 studies [10, 140] |
| | *Disposing of materials*, n = 9 studies [28, 47, 59, 95, 105, 112, 120, 122, 132, 135] | *Disposing of materials*, n = 10 studies [24, 32, 38, 42, 80, 81, 88, 102, 110, 116, 117, 138] | *Disposing of materials*, n = 2 studies [41, 50] |
| | *Obtaining materials*, n = 14 studies [47, 59, 67, 75, 76, 86, 91, 93–96, 99, 119–121, 126, 132, 135] | *Obtaining materials*, n = 18 studies [24, 37, 38, 48, 54, 56, 63, 69, 71, 78, 79, 81, 89, 106, 109, 110, 115–117, 138, 143] | *Obtaining materials*, n = 7 studies [8–10, 50, 64, 82, 83, 113] |
| | *Using/wearing materials*, n = 14 studies [28, 33, 47, 51, 59, 67, 68, 86, 91, 93–96, 105, 112, 114, 119, 120, 122, 123, 126, 135, 137] | *Using/wearing materials*, n = 26 studies [24, 25, 32, 36–38, 42, 44, 54, 56, 57, 60, 63, 69, 71, 72, 78–81, 84, 88, 89, 102, 106, 108–110, 115–118, 128, 138, 142] | *Using/wearing materials*, n = 9 studies [8–10, 41, 46, 49, 50, 64, 82, 83, 92, 104, 113, 136, 140] |
| | *Washing/drying materials*, n = 6 studies [28, 93–95, 105, 114, 132] | *Washing/drying materials*, n = 4 studies [37, 56, 81, 106, 108, 116, 117] | *Washing/drying materials*, n = 2 studies [10, 50] |
| **Perceptions of menstrual practices (n = 55 studies)** | n = 20 studies [28, 47, 51, 59, 67, 68, 86, 93–97, 99, 105, 112, 114, 119–123, 126, 132, 137] | n = 24 studies [24, 32, 36–38, 42, 54, 56, 57, 60, 63, 69, 71, 72, 78–81, 84, 88, 106, 108–110, 116, 117, 128, 138] | n = 11 studies [8–10, 41, 46, 49, 50, 64, 82, 92, 113, 136] |
| | *Bathing*, n = 2 studies [47, 105] | *Bathing*, n = 2 studies [56, 143] | *Bathing*, n = 0 studies |
| | *Disposing of materials*, n = 11 studies [28, 47, 51, 59, 105, 112, 114, 120–122, 132] | *Disposing of materials*, n = 7 studies [24, 25, 38, 42, 81, 88, 116, 117] | *Disposing of materials*, n = 2 studies [41, 50] |
| | *Obtaining materials*, n = 7 studies [67, 94, 96, 119–121, 132] | *Obtaining materials*, n = 8 studies [24, 37, 54, 56, 63, 78, 79, 81, 116] | *Obtaining materials*, n = 4 studies [8–10, 64, 82] |
| | *Using/wearing materials (all)*, n = 18 studies [28, 47, 51, 59, 68, 86, 93–96, 99, 105, 112, 114, 119–123, 126, 137] | *Using/wearing materials (all)*, n = 27 studies [24, 25, 27, 32, 36–38, 44, 54, 56, 57, 60, 63, 69, 71, 72, 78–81, 84, 88, 106, 108–110, 116, 117, 127, 128, 138] | *Using/wearing materials (all)*, n = 11 studies [8–10, 41, 46, 49, 50, 64, 82, 92, 113, 136] |
| | *Using/wearing tampons*, n = 6 studies [51, 68, 95, 99, 105, 121] | *Using/wearing tampons*, n = 15 studies [27, 32, 36, 37, 44, 56, 63, 69, 79–81, 106, 110, 116, 117, 127] | *Using/wearing tampons*, n = 6 studies [41, 49, 50, 64, 82, 113] |
| | *Washing/drying materials*, n = 2 studies [93, 94, 114] | *Washing/drying materials*, n = 5 studies [37, 56, 81, 106, 108, 116, 117] | *Washing/drying materials*, n = 1 study [50] |
| **Perceptions of physical environment (n = 18 studies)** | n = 8 studies [26, 28, 47, 51, 111, 122, 126, 132] | n = 9 studies [24, 25, 54, 80, 81, 88, 115, 117, 138] | n = 1 study [10] |

(Continued)

Table 3. (Continued)

| Theme | High trustworthiness (n = 36 studies) | Medium trustworthiness (n = 48 studies) | Low trustworthiness (n = 20 studies) |
|---|---|---|---|
| **Negative emotional responses (n = 87 studies)** | n = 33 studies [28, 33, 34, 40, 47, 51, 53, 58, 59, 61, 65, 67, 68, 73–77, 86, 91, 93–97, 99, 105, 111, 112, 114, 119–122, 124, 126, 132, 135, 137, 139] | n = 37 studies [24, 25, 27, 32, 36–38, 42, 44, 45, 48, 54–57, 60, 69–72, 78–81, 85, 87–90, 98, 100, 106, 108–110, 116–118, 125, 127, 128, 138, 143] | n = 17 studies [8–10, 35, 46, 49, 50, 62, 64, 82, 92, 101, 104, 113, 129, 136, 140, 141] |
| | *Bother*, n = 16 studies [47, 53, 58, 59, 65, 68, 75, 77, 94, 95, 105, 112, 119, 120, 122, 124, 135] | *Bother*, n = 18 studies [25, 36, 38, 54, 56, 69, 71, 72, 78, 80, 81, 87, 98, 106, 108–110, 117, 128, 138] | *Bother*, n = 7 studies [8, 35, 46, 49, 64, 140, 141] |
| | *Disgust*, n = 16 studies [28, 33, 47, 58, 68, 73, 74, 86, 93, 94, 96, 97, 99, 111, 114, 119–121, 132, 135] | *Disgust*, n = 15 studies [32, 38, 54–56, 60, 70, 79, 81, 87, 98, 110, 117, 138, 143] | *Disgust*, n = 3 studies [64, 92, 136] |
| | *Shame and distress*, n = 28 studies [28, 33, 34, 40, 47, 51, 53, 58, 59, 61, 67, 68, 75, 76, 86, 93–97, 99, 105, 111, 112, 114, 119–121, 132, 135, 137, 139] | *Shame and distress*, n = 27 studies [24, 27, 32, 38, 44, 48, 54–56, 69, 71, 72, 78–81, 85, 87–90, 98, 100, 106, 108–110, 116, 117, 125, 127, 128, 143] | *Shame and distress*, n = 13 studies [8, 9, 46, 50, 64, 82, 101, 104, 113, 129, 136, 140, 141] |
| **Positive emotional responses (n = 42 studies)** | n = 14 studies [26, 33, 40, 47, 68, 73–75, 93–97, 105, 114, 119, 120, 135] | n = 19 studies [27, 32, 37, 43, 54, 56, 60, 63, 69–71, 78, 80, 81, 85, 88, 98, 106, 143] | n = 9 studies [46, 49, 62, 64, 83, 104, 129, 136, 141] |
| | *Pride*, n = 0 studies | *Pride*, n = 4 studies [32, 78, 106, 143] | *Pride*, n = 1 study [136] |
| | *Relief*, n = 11 studies [26, 40, 47, 68, 75, 93, 94, 96, 105, 114, 119, 120, 135] | *Relief*, n = 12 studies [27, 32, 43, 56, 60, 69–71, 80, 85, 98, 106] | *Relief*, n = 7 studies [46, 49, 62, 64, 104, 129, 141] |
| **Confidence to engage in other activities during menstruation (n = 16 studies)** | n = 9 studies [28, 33, 40, 47, 73, 112, 114, 119, 122] | n = 7 studies [38, 56, 80, 81, 106, 108, 117, 118] | n = 0 studies |
| **Individual menstrual factors (n = 71 studies)** | n = 29 studies [26, 28, 33, 34, 39, 40, 47, 51–53, 58, 59, 61, 65, 67, 73, 75–77, 93, 95, 96, 99, 105, 112–114, 119–122, 124, 135, 137] | n = 32 studies [24, 25, 27, 32, 36–38, 54–56, 60, 63, 69, 71, 72, 78, 80, 81, 85, 87, 100, 102, 103, 106, 107, 109, 110, 116, 127, 128, 133, 134, 138, 142] | n = 10 studies [8–10, 35, 41, 46, 49, 62, 64, 129, 140] |
| | *Fatigue*, n = 13 studies [33, 34, 40, 51, 61, 76, 77, 111, 114, 119, 120, 122, 137] | *Fatigue*, n = 8 studies [25, 56, 72, 79, 80, 87, 100, 138] | *Fatigue*, n = 3 studies [9, 41, 140] |
| | *Gastrological*, n = 10 studies [40, 51, 53, 58, 61, 75, 76, 114, 128, 135] | *Gastrological*, n = 7 studies [55, 56, 71, 80, 106, 128, 134] | *Gastrological*, n = 3 studies [8, 49, 140] |
| | *Neurological*, n = 10 studies [51, 53, 61, 75, 76, 114, 119, 120, 122, 128] | *Neurological*, n = 9 studies [32, 55, 56, 71, 72, 78, 80, 106, 128] | *Neurological*, n = 2 studies [8, 41] |
| | *Pain*, n = 29 studies [26, 28, 33, 34, 39, 40, 47, 51–53, 58, 59, 61, 65, 67, 73, 75–77, 93, 95, 96, 99, 105, 112, 114, 119–122, 124, 135, 137] | *Pain*, n = 31 studies [24, 25, 27, 32, 36–38, 54–56, 60, 63, 69, 71, 72, 78–81, 85, 87, 102, 103, 106, 107, 109, 110, 116, 127, 128, 133, 134, 138, 142] | *Pain*, n = 9 studies [8–10, 35, 41, 46, 49, 62, 64, 129] |
| **Pain management (n = 47 studies)** | n = 24 studies [26, 28, 33, 34, 40, 47, 51–53, 58, 59, 61, 65, 67, 73, 75, 77, 93, 105, 112, 114, 119, 120, 122, 137] | n = 18 studies [32, 38, 44, 55, 56, 63, 71, 79–81, 102, 106, 116, 127, 128, 133, 138, 142] | n = 5 studies [8–10, 41, 64, 129] |
| | *Perceptions of pain management*, n = 15 studies [33, 34, 40, 47, 51, 53, 58, 61, 67, 73, 77, 105, 119, 120, 122] | *Perceptions of pain management*, n = 12 studies [32, 44, 55, 56, 71, 80, 81, 102, 106, 116, 128, 133] | *Perceptions of pain management*, n = 3 studies [8, 64, 129] |
| | *Use of:* | *Use of:* | *Use of:* |
| | *Alcohol*, n = 1 study [93] | *Alcohol*, n = 2 studies [32, 106] | *Alcohol*, n = 0 studies |
| | *Heat*, n = 9 studies [34, 40, 52, 53, 58, 59, 61, 120, 122] | *Heat*, n = 4 studies [34, 56, 71, 80, 142] | *Heat*, n = 0 studies |
| | *Herbal remedies*, n = 5 studies [40, 65, 67, 105, 122] | *Herbal remedies*, n = 1 study [55] | *Herbal remedies*, n = 1 study [41] |
| | *Hormonal contraceptives*, n = 2 studies [51, 58, 61, 77, 105, 120] | *Hormonal contraceptives*, n = 5 studies [56, 80, 81, 116, 133] | *Hormonal contraceptives*, n = 4 studies [8, 10, 64, 129] |
| | *Analgesic or anti-inflammatory medicines*, n = 17 studies [26, 34, 40, 47, 51–53, 58, 61, 67, 75, 77, 112, 114, 120, 122, 137] | *Analgesic or anti-inflammatory medicines*, n = 15 studies [32, 38, 56, 63, 71, 79–81, 102, 106, 127, 128, 133, 138, 142] | *Analgesic or anti-inflammatory medicines*, n = 2 studies [64, 129] |
| | *Sex*, n = 2 studies [33, 73] | *Sex*, n = 1 study [56] | *Sex*, n = 1 study [41] |
| **IMPACTS** | | | |

*(Continued)*

Table 3. (Continued)

| Theme | High trustworthiness (n = 36 studies) | Medium trustworthiness (n = 48 studies) | Low trustworthiness (n = 20 studies) |
|---|---|---|---|
| **Mental burden (n = 36 studies)** | n = 14 studies [40, 47, 51, 53, 61, 77, 95, 112, 119, 120, 122, 126, 132, 137] | n = 22 studies [25, 36, 38, 55, 56, 69–71, 78–81, 88, 98, 103, 106, 109, 117, 118, 128, 134, 138, 143] | n = 4 studies [46, 49, 64, 129] |
| **Participation (n = 72 studies)** | n = 25 studies [28, 34, 40, 47, 51–53, 58, 61, 65, 67, 76, 77, 86, 93–96, 99, 105, 111, 112, 114, 119, 120, 122, 135, 137] | n = 35 studies [25, 27, 32, 36–38, 44, 45, 48, 54–57, 60, 63, 71, 72, 79–81, 84, 87, 98, 100, 103, 106–110, 116–118, 127, 128, 134, 138, 143] | n = 12 studies [8, 41, 49, 50, 64, 92, 101, 113, 129–131, 136, 141] |
| | *Participate but would prefer not to,* n = 9 studies [40, 47, 53, 58, 61, 77, 105, 112, 122] | *Participate but would prefer not to,* n = 9 studies [56, 63, 80, 98, 100, 106, 107, 109, 128, 138] | *Participate but would prefer not to,* n = 2 studies [8, 64] |
| | *Employment is impacted,* n = 10 studies [34, 47, 53, 61, 76, 105, 111, 112, 114, 119, 122] | *Employment is impacted,* n = 7 studies [34, 56, 57, 80, 100, 103, 106, 108, 118] | *Employment is impacted,* n = 2 studies [64, 129] |
| | *Participation in physical activities is impacted,* n = 10 studies [34, 40, 47, 51, 65, 77, 93–95, 99, 105, 112] | *Participation in physical activities is impacted,* n = 19 studies [27, 32, 36, 38, 44, 45, 54, 56, 60, 63, 71, 72, 80, 81, 87, 106, 108, 109, 116, 127] | *Participation in physical activities is impacted,* n = 6 studies [8, 41, 49, 50, 113, 141] |
| | *Do not participate in 'cold' activities,* n = 3 studies [65, 94, 95, 105] | *Do not participate in 'cold' activities,* n = 7 studies [45, 56, 63, 71, 81, 84, 106, 108] | *Do not participate in 'cold' activities,* n = 2 studies [41, 50] |
| | *Do not participate in religious practices,* n = 3 studies [67, 86, 120] | *Do not participate in religious practices,* n = 2 studies [60, 63] | *Do not participate in religious practices,* n = 2 studies [8, 92] |
| | *Do not participate in education,* n = 6 studies [34, 47, 58, 61, 65, 122] | *Do not participate in education,* n = 12 studies [32, 36, 48, 56, 63, 71, 80, 87, 103, 107, 127, 143] | *Do not participate in education,* n = 5 studies [8, 49, 64, 101, 129] |
| | *Do not participate in social activities,* n = 11 studies [34, 40, 58, 61, 65, 93–95, 99, 112, 119, 122, 135] | *Do not participate in social activities,* n = 10 studies [36, 38, 45, 48, 71, 72, 80, 81, 87, 106] | *Do not participate in social activities,* n = 3 studies [41, 92, 129] |
| **Relationships (n = 37 studies)** | n = 15 studies [33, 34, 40, 47, 61, 67, 73, 74, 86, 93, 105, 112, 119, 122, 132, 137] | n = 16 studies [24, 25, 32, 38, 55, 56, 60, 69, 71, 78, 80, 81, 88, 100, 106, 116] | n = 6 studies [49, 64, 101, 104, 129, 140] |
| | *Intimate partner,* n = 12 studies [33, 47, 61, 67, 73, 74, 86, 93, 105, 112, 119, 122, 137] | *Intimate partner,* n = 7 studies [38, 56, 60, 80, 81, 100, 116] | *Intimate partner,* n = 2 studies [101, 104] |
| | *Friends/family/colleagues,* n = 7 studies [34, 40, 47, 61, 93, 119, 132] | *Friends/family/colleagues,* n = 10 studies [24, 25, 32, 55, 56, 69, 71, 78, 88, 106] | *Friends/family/colleagues,* n = 4 studies [49, 64, 129, 140] |

Italicised terms provide citations for sub-themes within the overarching themes in the first column.

Box 1. Illustrative quotations for antecedents of menstrual experiences

### Socio-cultural context

"Some women articulated common cultural assumptions about menstruation as frankly unattractive, using a strong language of disgust and repulsion," [pg. 165, 73]

"Women reported feeling shameful and embarrassed about their menstruating bodies" [pg. 1483, 86]

"Participants wrote about the deep sense of shame associated with menstrual bleeding" [pg. 619, 97].

"[at menarche] I was happy because I was growing up" [pg. 1336, 96]

"My oldest sister always told me, you've got to act like a lady [after menarche] and that just crushed me because I was having a good time, you know" [pg. 101, 95].

"Periods made him [transgender menstruator] aware that he embodied the "wrong body" and "for me, the issue has been that it (menstruation) hasn't been corresponding to who I am"" [pg. 44, 28]

### Behavioural expectations

"Your mother taught ya: you keep it to yourself, you keep yourself clean" [pg. 39, 38]

"It [menstruation] shouldn't kind of show, it should be, girls should have it, but it shouldn't appear as if they do" [pg. 607, 47]

"I knew I had to keep this to myself and I was horrified that someone might know what was happening" [pg. 621, 97]

"One time when I was visiting my sister I was having a heavy period and bled through onto the bed and she's like 'Oh my god! That's so gross.' I felt ashamed" [pg. 9, 75]

### Social support

"My mum knew about it [menstruation], but didn't tell me anything about it, and all she said was: 'Oh, it just shows that you're grown up now . . . but you'll have these once a month, until you are about 50', and that was it" [pg. 401, 109].

"One time it [menstrual fluid] went through my jeans. My best friend told me and I had to take her shirt and run to the bathroom" [pg. 126, 56]

"I had female soldiers in my platoon, and we—it was never an issue . . . we could share whatever we needed [menstrual materials], type of thing" [pg. 345, 138].

"I was in the office and he [her boss] saw me and he asked what was going on and, and I told him a little and basically the, the beginning of the conversation ended very quickly when I said, "I have my period," and he said, "Okay, you're not feeling well. I know you have a problem, go home. And if you feel better and you feel up to it, come back"" [pg. 72, 129].

"A friend of mine used to say and you know because I would say or I am not feeling very well [dysmenorrhea] so I wasn't able to do something and she would get very impatient and say "Oh god we all have our period, you know, you are not unique" and it used to frustrate her that I wouldn't be able to do things like dragon boat or whatever because I wasn't feeling very well and I used to feel quite guilty and, you know, like I was making a bigger deal of it than perhaps it was, but I actually was feeling ill" [pg. 231, 40]

"She's wearing the rag" [statement by adolescent boy, pg. 19, 81]

"Geez, you smell like dead fish!" [statement by adolescent boy, pg. 42, 32]

"I felt like doctors are telling you it's in your head" [pg. 7, 125]

### Knowledge

"When it started [menarche], I was scared to death, thought I was dying" [pg. 117, 95]

"My mom should've taught me how you take care of a pad and where you put it before my period actually started. . . I thought you could just stick it in the toilet, and it got stuck in the toilet, and one of the camp leaders was like, "Who put this in here?" I was so embarrassed" [pg. 1299, 135].

### Resource limitations

"It's [lacking the funds to purchase menstrual materials] horrible, it's degrading. Nobody should have to be like that" [pg. 11, 48]

"I [trans- or non-binary person who menstruates] only rarely use men's restrooms, as I have a lot of anxiety about doing so around strangers" [pg. 1244, 24].

---

### Box 2. Illustrative quotations for menstrual experiences

### Menstrual practices

"Yesterday I decided I was going to wear pads for this period. . . I just felt absolutely revolting, I was all sticky and I had to give myself a bit of a douche. . . I just find it uncomfortable and messy and it gets a bit pongy" [pg. 22, 92]

"I feared I'd smell or spot my clothes and then everyone—especially boys—would know I'd "had the curse." The bulge of those old type Kotex I was sure told everyone what was happening" [pg. 129, 95].

### Perceptions of menstrual practices

"I didn't like it in the dorm bathroom before they put the trash can in one of the stalls. Like I was really upset that we had to bring it out to the main trash can. I thought that was awful" [pg. 389, 88].

"I don't mind going into a chemist or 'um, like in town or something like that. But I used to live in [a nearby village]. There was a little local shop opposite and there was a

man in there, and he was quite old and I had to go and buy some. I was really embarrassed. It was silly but I felt really sort of oh gosh!" [pg. 156, 82].

### Negative emotional responses

"I feel that I have passed that point in my life so I don't actually need my periods to continue and I just see them as just a blight really now" [pg. 94–95, 119]

"that's when I got my first period . . . I was really ashamed" [pg. 671, 139]

"I was kind of ashamed of it [menarche]. I thought there was something wrong with me" [pg. 92, 94].

"[Menarche was] the most horrifying and disgusting thing in my life" [pg. 619, 97]

### Positive emotional responses

"I always say when I get my period "welcome!" and feel physically and mentally clean when it arrives" [pg. 32, 105].

### Confidence to engage in activities during menstruation

"It was freedom. It freed us to do more sports. . . with the new products [tampons] I felt a lot more secure" [pg. 86, 56].

"Avoidance coping mechanisms and behaviours . . . rescheduling activities at home and at work, cancelling meetings both social and professional" [pg. 80, 119]

### Individual menstrual factors

"[Participants] spoke of feeling reluctant to complain about menstrual symptoms because 'everyone gets them'" [pg. 251, 122]

### Box 3. Illustrative quotations for the impacts of menstrual experiences

### Mental burden

"Women experience that menstruation and menstrual symptoms affect their activities and performance in daily life. For instance, they experience how their concentration, patience, and efficiency at work or in school are reduced, which leads to feelings of not being as productive as expected in the social context" [pg. 608, 47]

"Never bleed[ing] again ever, ever, ever, ever and that's the biggest one, that's it. Everything else stems from that, lack of energy, feeling unclean, having to worry about where I am going on certain days; that it will all just stop" [pg. 94, 119]

### Participation

"People say to her "If [I am] not doing something: 'Why are you not doing that, you are a woman, you should be able to deal with that'"" [pg. 39, 8]

"I've had very few days off work due to endometriosis purely because I just go and I'm stubborn. I'll just go into work no matter what you know. I probably shouldn't but I do and this they're making redundancies and things at my work so you do what you can to get your to keep your job" [pg. 40, 61].

### Relationships

"A wonderful secret that they [boys] would never know about and never experience" [pg. 138, 32]

"I want to talk to my butch friends about my period, but even though I'm ok with it, I'm afraid it will be triggering for them so I don't, which can feel isolating" [pg. 1246, 24].

be breaking a social norm [8, 28, 38, 47, 48, 51, 56, 59, 64, 65, 67, 68, 77, 81, 87–89, 93, 95, 96, 106–108, 110–112, 116, 119–122, 125, 133, 139].

There were a few examples in the studies from participants who reached menarche from the late 20th C onwards of insisting that this secrecy should be challenged, for example, displaying menstrual materials publicly [pg. 54, 81] or insisting on menstrual discussions with boys and men who did not want to engage [pg. 110, 81, pg. 8, 110]. However, it was also noted by authors that "women who began menstruating in the 1980s and 1990s told me that they felt no shame or discomfort about their monthly cycles. They were dubious as to whether menstrual taboos still operated in Australia. Yet their memories of managing bleeding reveal a continuing expectation that menstruation be masked as much as possible" [pg. 244, 117] and "although some participants who began menstruating in the late 20th and early 21st Cs specifically questioned this norm "as they feel that menstruation is a natural phenomenon that all women experience and should therefore not have to be concealed," they, "still want to hide their periods, keep them private, and choose not to confront the social norm"" [pg. 608, 47].

**Social support.** The perception of social support (or a lack of) from family members, friends and work colleagues, adolescent boys, and healthcare workers, influenced participants' experiences of menstruation and its impact on their lives [8–10, 24–28, 32–38, 40, 42, 43, 46–51, 54–67, 69–73, 75–85, 87–90, 93–97, 99–106, 108–122, 124, 125, 127–129, 135–138, 140, 141, 143].

Mothers were the family member most spoken about by participants, normally with regards to menarche [8, 25, 32, 34–36, 38, 42, 43, 47, 49, 50, 54, 56, 57, 61–65, 69–72, 76, 79, 81–84, 88, 93–96, 99, 102, 104, 106–110, 113, 114, 116, 119–122, 125, 127–129, 135, 140, 141]. Adolescents felt the emotional support offered by their mothers at menarche was mixed, from appreciating them throwing a menarche party [43] to "seem(ing) upset like she had to deal with something she didn't want to" [pg. 143, 56]. In general, emotionally supportive mothers were appreciated by participants [32, 34, 38, 43, 47, 60, 61, 64, 79, 81, 96, 106, 114, 127, 135, 144] and emotionally unsupportive mothers criticised [8, 32, 38, 49, 56, 63, 64, 70, 81, 96, 106, 107, 109, 110, 119, 128]. Several participants specifically stated that they had received neither positive nor negative support; their mother dealt with menarche matter-of-factly and it was rarely mentioned afterwards. These participants felt they could have been better prepared for the experiences of

menstruation [32, 38, 56, 63, 65, 70, 95, 104, 106, 109, 110, 119]. Several studies included participants stating that they appreciated the social support they received during adolescence from other family members, particularly grandmothers, sisters and aunts [65, 71, 99, 106, 110, 119, 121, 122, 127, 135, 144], and sometimes brothers and fathers [32, 43, 54, 61, 71]. A few participants stated that they had received inadequate support from their fathers [32, 56, 61, 95, 121].

Friends and colleagues were often mentioned as assisting menstruators with concealing and containing menstruation across the lifespan through providing menstrual materials, notifying one another of stains, or helping hide menstrual practices [24, 37, 43, 47, 48, 56, 63, 75, 78, 88, 94, 95, 112, 115, 118, 122, 124, 138]. Some participants, particularly those with menstrual disorders or discomfort, reported having emotionally and/or practically supportive work colleagues, who allowed them flexibility in the workplace. These were often women [47, 80, 112, 114, 115, 119, 122, 124, 129, 138], although some participants specifically mentioned supportive men [80, 118, 124, 129]. Several participants indicated that other women became frustrated with them when they were unable to fulfil social or work duties due to debilitating pain or fatigue, often implying that menstrual discomforts were a 'normal' part of life and should not impact on participation or work quality [8, 40, 61, 71, 76, 80, 103, 119, 122, 128].

Many participants described incidences of bullying by boys during adolescence [32, 35, 54, 78, 81, 88–90, 93, 96, 97, 99, 110, 117, 141, 143]. These included being teased for using or possessing menstrual materials or because they 'smelled', leading to embarrassment [32, 35, 54, 78, 81, 89, 90, 96, 97, 143]. Bullying was still evident in studies where participants began menstruating in the early 21st C [54, 78, 88–90, 96, 97, 99, 110, 114, 121, 127, 141, 143].

In several studies participants with menstrual disorders (e.g., endometriosis, menorrhagia, dysmenorrhea) had previously consulted healthcare workers and had their concerns dismissed [26, 40, 55, 57, 58, 61, 71, 75, 85, 99, 101, 103, 111, 119]. Where healthcare workers did acknowledge the experience of pain and/or heavy bleeding they often expressed that menstrual symptoms were just a normal part of being a woman, or that the patient must have "a very low pain threshold" [pg. 45, 71]. They either could not do anything for them or recommended painkillers [40, 55, 57, 58, 61, 71, 80, 81, 85, 111, 112, 119, 122, 137]. Less frequently participants spoke about the relief of a positive interaction with a healthcare professional (generally following many negative experiences) where they "felt heard" [pg. 234, 40] and were sometimes assisted in developing pain management strategies they considered effective [40, 51, 55, 59, 61, 77, 81, 85, 99, 111, 122], leading to improved impacts on mental burden, relationships and participation. A few participants noted that their intimate partner was supportive of them when they were experiencing menstrual pain [56, 61, 81, 119, 122].

**Knowledge.** Participants' often believed that they lacked sufficient, accurate knowledge about the biology of menstruation [8, 32, 35, 36, 38, 39, 41, 42, 45, 50, 55–60, 63–65, 67, 69–72, 74, 75, 81, 85, 87, 89, 91, 93–97, 102, 104–106, 109, 110, 114, 116, 119–122, 127, 128, 130, 135, 136, 141], how it is linked to reproduction [32, 33, 41, 56, 58, 60, 63, 71, 74, 78, 81, 95, 106, 120] and how to physically manage menses [8, 32, 36, 38, 39, 42, 43, 47, 50, 51, 56, 58, 63, 69, 71, 72, 81, 86, 88, 91, 94–96, 99, 104–106, 110, 114, 116, 117, 119–122, 127, 135, 140], particularly at menarche or during adolescence.

Participants who had reached menarche in the early-mid 20th C often indicated that they had not known what menstruation was at the time of their menarche [8, 38, 42, 56, 60, 65, 69, 72, 81, 87, 89, 93–97, 102, 109, 110, 116, 119, 120, 135]; in most cases this lack of knowledge led to distress when first bleeding was discovered. In the early 20th C, adolescent participants were often instructed not to swim in cold water or wet their hair because they were incorrectly advised that it was physically dangerous to do so during menstrual bleeding [45, 60, 63, 65, 71, 81, 95].

Across the timespan of studies there was limited discussion of being taught about menstruation in a formal school environment; some participants reported that at menarche their mother explained some of the biological and reproductive aspects of menstruation to them [32, 38, 42, 56, 63, 64, 72, 81, 95, 96, 106, 109, 116, 119, 120, 135]. Many participants were educated on the physical management of menstruation by their mothers, at the time of their menarche [32, 38, 43, 63, 88, 93–96, 106, 116, 127]. However, several noted that the information their mother gave them was inadequate and they would have appreciated more instruction [32, 36, 38, 47, 50, 56, 91, 94, 95, 104, 110, 119, 135, 138].

**Resource limitations.**   Several participants indicated resource limitations related to sourcing menstrual materials [8–10, 37, 40, 48, 56, 81, 98, 106, 114, 117, 119, 120, 126, 143] or having access to an adequate place to regularly change menstrual materials, clean themselves during menstruation, or dispose of menstrual materials [10, 25, 26, 28, 47, 51, 54, 78, 80, 81, 88, 94, 106, 108, 111, 115, 117, 120, 122, 126, 132, 138]. Resource limitations were most prominent for participants receiving low incomes or who were part of marginalised, lower-socioeconomic groups [9, 10, 37, 48, 62, 65, 126, 132], sometimes also experiencing homelessness [9, 10, 37, 126, 132]. These sub-populations have often not been included in qualitative studies of menstrual experiences in HICs, with the earliest study to purposefully recruit low income participants being published in 2007 [62], and those experiencing homelessness in 2017 [9]. For non-binary or transgender individuals, access to an adequate place to manage their menstruation was often determined by the societal norm that menstrual disposal options are not available in most men's toilets [24–26, 28]. Similarly to low income populations, the experiences of trans- and non-binary people who menstruate have not traditionally been researched, with the first study which purposively sampled for such participants being published in 2016 [24]. Across the studies and the review timespan, resource limitations often led to shame and distress, [24, 47, 48, 78, 80, 81, 88, 108, 112].

## Menstrual experience

**Menstrual practices.**   The practices used to contain and clean menstrual bleeding varied over time, from the use of washable cloths and menstrual belts through to disposable adhesive pads and tampons and later reusable menstrual cups [8–10, 24, 25, 28, 32, 33, 36–38, 41, 42, 44, 46, 47, 49–51, 54, 56, 57, 59, 60, 63, 64, 67–69, 71, 72, 78–84, 86, 88, 89, 91–96, 102, 104–106, 108–110, 112–120, 122, 123, 126, 128, 135–138, 140, 142]. Some menstrual practices impacted on respondents' confidence or choice of whether to engage in other activities whilst menstruating, particularly with regards to whether they swam or engaged in sporting activities [38, 44, 56, 81, 95, 106, 109]. Where participants had negative perceptions of their menstrual practices this led to negative emotions such as distress, disgust and embarrassment [24, 25, 32, 37, 38, 44, 46, 48, 54, 56, 59, 67–69, 71, 72, 79, 81, 82, 88, 92, 94, 95, 110, 112, 114, 116, 117, 120, 126, 135, 136].

Participants often selected the material they used based on absorbance or how long it could be used for without needing to change [57, 59, 81, 114, 119, 120, 122, 123, 126, 128]. Participants with heavy bleeding particularly needed to plan their menstrual practices, often through wearing tampons and pads, or multiple pads, at the same time [53, 57, 61, 69, 81, 112, 119, 120, 122, 123]. Across the timespan of studies many participants expressed discomfort at using pads, describing them as hot and abrasive [38, 56, 59, 71, 72, 81, 92, 95, 108, 117]. Some non-binary or trans-men who menstruate preferred not to use tampons or menstrual cups because the insertion of these products contributed to their gender dysphoria [25, 27, 28].

**Perceptions of menstrual practices.**   In many studies participants shared their appraisal and decision making around different menstrual practices [8–10, 24, 28, 32, 36–38, 41, 42, 46,

47, 49–51, 54, 56, 57, 59, 60, 63, 64, 67–69, 71, 72, 78–82, 84, 86, 88, 92–97, 99, 105, 106, 108–110, 112–114, 116, 117, 119–123, 126, 128, 132, 136–138]. Such choices were normally based on the availability of materials, facilities and services, their personal preferences (e.g., how often a material would need changing), the expectations placed on them by themselves and others and their own personal needs. This was frequently dictated by the priority to conceal menstrual status, although for some non-binary and transgender menstruators this concealment was also related to their desire to 'pass' as a cis-man [24, 25]. Containing menstrual fluid and concealing one's menstrual status was described as difficult in the early 20th C (i.e., when rags and menstrual belts were the norm) but became easier in the mid-late 20th C (i.e., once tampons, slimmer adhesive pads and menstrual cups became available) [25, 38, 51, 56, 63, 71, 81, 82, 94, 95, 106, 114, 117, 136].

The choice of disposal practices was normally based on whether it would conceal the users' menstrual status [24, 25, 28, 41, 42, 47, 50, 51, 81, 114, 116, 117, 120, 132], and where disposal did not conceal menstrual status the participant sometimes felt intense shame [38, 88, 112, 122]. Washing reusable cloths was discussed in some publications where participants reached menarche in the early 20th C, and it was repeatedly noted that this "burdensome" and "distasteful" [pg. 238, 117] chore must be done discreetly, even though there was often no way of doing so, for example, where washed cloths needed to hang on shared clotheslines [81, 93, 94, 106, 108, 116, 117]. Across the timespan of studies, participants often tried to hide menstrual materials when purchasing them and described embarrassment when the cashier was a man [10, 24, 37, 54, 56, 67, 78, 79, 81, 82, 93, 116, 119, 120]. Participants who could not afford to purchase menstrual material often felt embarrassed at having to obtain them from friends or non-profit organisations such as shelters [9, 10, 37, 48].

Several studies specifically discussed participants' perceptions of tampon use. Many participants who reached menarche in the mid-20th C were told by their mothers that they could, or should, not use tampons [32, 49, 50, 63, 81, 106, 110, 121], at menarche they were either expressly forbidden without grounds, or told that tampon use would result in them having 'lost' their virginity or contribute to them engaging in sexual activity. Some participants believed this when they were told, but appeared to have changed their mind by the time they engaged in the studies (data collected from the late 20th C onwards) [56, 81, 116, 117]. Many participants who used tampons spoke of them as being "liberating" or "emancipatory" [pg. 242, 117] as they were easy to conceal and allowed them to partake in activities which they could not previously [49, 50, 56, 81, 106, 117]. Where participants reached menarche in the mid-20th C onwards there was concern around the risk of toxic shock syndrome when using tampons, sometimes leading to a reluctance to try them [36, 56, 69, 80, 81, 127].

**Perceptions of physical environments.** Participants often noted that they did not believe that they had access to private, appropriate facilities where they could regularly bathe, dispose of menstrual materials and change their menstrual materials [10, 24–26, 28, 47, 51, 54, 80, 81, 88, 111, 115, 117, 122, 126, 132, 138]. This caused particular distress for those with heavy bleeding [10, 24, 47, 80, 81, 88, 115, 117, 122, 138] and those who identified as non-binary or transgender and felt uncomfortable using 'men's' rooms when menstruating, for fear of being identified as a non cis-man, which they believed could be dangerous for them [24–26, 28].

**Negative emotional responses.** Participants described a variety of negative emotional responses as part of their menstrual experience. Hennegan *et al.*'s review of LMIC studies identified 'shame and distress' as a theme and part of the integrated model, with positive emotions described as divergent cases under this theme [15]. In contrast, studies included in our review reported a wider array of emotional reactions to menstruation. Less intense negative responses were often described, such as feeling menstruation was inconvenient or bothersome. Further, in inductively coding study findings we identified different antecedents and impacts of

negative and positive emotional responses and so separated these to capture experiences reported in HIC study populations.

Menstruation was often considered a bother, generally with regards to coping with physical symptoms, menstrual practices and the expectation of containment and concealment [8, 25, 32, 35, 36, 38, 40, 46, 47, 49, 53–59, 64, 65, 68, 69, 71, 73, 75, 77–81, 85, 87, 94, 95, 98, 105, 108–110, 114, 117, 119, 120, 122, 138, 140, 141], or being shameful, worrying or distressing [8, 9, 24, 27, 28, 32–34, 36, 38, 42, 44–48, 50, 51, 53–56, 58–61, 64, 65, 67–69, 71–78, 80–82, 85–90, 93–101, 104, 105, 108–114, 116, 117, 119–122, 125, 127–129, 132, 135, 136, 139–141, 143] particularly at menarche [8, 27, 28, 32, 47, 56, 71, 81, 88–90, 93, 94, 96, 97, 109, 110, 113, 116, 119, 121, 127, 135, 136, 139–141]. Menstruation also often made participants feel disgusting or unclean [28, 33, 38, 47, 54–56, 58, 60, 64, 68, 73, 74, 81, 86, 87, 92–94, 96–99, 110, 111, 117, 119, 121, 135, 136, 143]. Negative emotional responses, particularly shame and worry, often led to increased mental burden [25, 34, 38, 46, 47, 51, 53, 55, 64, 77, 81, 88, 109, 112, 117–120, 122, 128, 132, 138, 143] and sometimes non-participation where the negative emotional responses were linked to concerns about respondents' own ability to conceal their menstrual status [48, 49, 55, 71, 93, 94, 119, 127, 135, 141].

**Positive emotional responses.** Positive emotional responses were reported by participants half as often as negative emotional responses [26, 27, 32, 33, 37, 40, 43, 46, 47, 49, 54, 56, 60, 62–64, 68–71, 73–75, 78, 80, 81, 83, 85, 88, 93–98, 104–106, 114, 119, 120, 129, 135, 136, 141, 143]. Positive emotional responses were commonly related to menarche, where participants were 'proud' to 'become a woman' [32, 43, 49, 56, 60, 63, 64, 71, 88, 93–97, 105, 106, 120, 135, 136, 144]. Occasionally participants mentioned relief on beginning their period each month, as this indicated that they did not have any menstrual disorders or were not pregnant [26, 32, 47, 68, 75, 98, 105]. Some participants noted experiencing pleasure during menstrual sex [33, 56, 73, 74]. Positive emotional responses often had beneficial impacts on participants' relationships, explored more below.

**Confidence to engage in activities during menstruation.** A number of studies included participants who spoke of their confidence (or lack of) to engage in activities during menstruation [28, 33, 38, 40, 47, 56, 73, 80, 81, 106, 108, 112, 114, 117–119, 122]. Often those who had begun using tampons or a menstrual cup particularly mentioned how this gave them the confidence to engage in more activities [28, 38, 56, 81, 114, 117]. Conversely, some participants who were ashamed of their menstrual status or experienced painful symptoms lacked the confidence to participate in activities during their period [38, 47, 56, 108, 119].

**Individual menstrual factors.** Across studies, individual participants reported physical symptoms accompanying their menstrual period. These varied in intensity from a clinically diagnosed menstrual, hormonal, or uterine bleeding disorder, to sub-clinical experiences (e.g., pain, fatigue and gastrological and neurological symptoms) reported by participants. The extent to which an individual experienced symptoms was integral to their menstrual experience in the context of the described antecedents, including their knowledge, access to support, and behavioural expectations to conceal or share experiences. Pain in particular contributed to an increased mental burden during menstruation [38, 40, 47, 53, 56, 61, 64, 80, 112, 129, 137, 145], and impacted on their relationships [40, 47, 55, 56, 61, 80, 105, 119, 129] and their participation in activities [8, 25, 28, 32, 34, 38, 40, 49, 51–53, 55, 56, 58, 61, 63–65, 71, 72, 77, 80, 81, 87, 103, 106, 107, 109, 112, 114, 119, 122, 127–129].

**Pain management.** Many participants indicated that they used pain management during menstruation, with varying success [8–10, 26, 28, 32–34, 38, 40, 41, 44, 47, 51–53, 55, 56, 58, 59, 61, 63–65, 67, 71, 73, 75, 77, 79–81, 93, 102, 105, 106, 112, 114, 116, 119, 120, 122, 127–129, 133, 137, 138, 142]. Many worried that medicines and hormonal contraceptives were bad for their body as they were "unnatural" [pg. 1340, 53]. Some refused to use these pain

management strategies for this reason [32, 51, 58, 71, 81, 105, 120, 133], others used them but expressed concern [32, 53, 64, 122].

## Impacts of menstrual experience

**Mental burden.**   Many participants described a significant mental burden, sometimes all month long, related to managing menstrual bleeding and experiencing physical symptoms, distress and bother [25, 38, 40, 46, 47, 49, 51, 53, 55, 56, 61, 64, 69, 71, 77–81, 88, 95, 98, 103, 109, 112, 117, 120, 122, 126, 128, 129, 132, 134, 137, 138, 143]. The mental burden was often attributed to "making sure" [pg. 18, 118] they were always concealing their menstrual status [38, 46, 47, 51, 53, 55, 64, 81, 88, 109, 112, 117–120, 122, 128, 132, 138, 143], as well as participating despite pain due to the assumption that menstrual suffering is normal [36, 38, 40, 47, 64, 71, 145]. The mental burden increased where participants experienced physical symptoms, including irregular periods, [38, 40, 47, 53, 56, 61, 64, 80, 103, 112, 129, 137, 145], or when there was uncertainty in whether a toilet would be available for menstrual management [47, 78, 80, 122, 126, 132]. Sometimes the mental burden was reduced by successful medical intervention to reduce pain or regulate periods [38, 47, 81, 112, 119].

**Participation.**   Participation in a variety of activities differed over time and between individual participants. Sometimes respondents did not participate in activities because they lacked the confidence that their menstrual practices would conceal their menstrual status, occasionally due to a past, embarrassing experience of failing to contain menstrual fluid and conceal physical symptoms [40, 48, 49, 55, 57, 58, 61, 71, 72, 80, 96, 98, 105, 112, 119, 122, 129, 134, 135, 137]. Other times, particularly in studies where participants reached menarche from the mid-20th C onwards, respondents were forced by mothers to go to school and participate in other activities during menstruation, even if they did not want to [8, 63, 64, 109, 110, 128]. Sometimes individuals chose to participate in activities despite menstruation, just "'getting on' with life" [pg. 1337, 53], particularly from the late 20th C onwards [38, 40, 47, 50, 53, 56, 110]. However, many who participated in activities would have preferred not to, but did so due to the behavioural expectation that menstruation should not stop them [8, 40, 47, 53, 56, 58, 61, 63, 64, 77, 80, 98, 100, 105, 107, 109, 112, 122, 128, 138]. During participation, these individuals often experienced distress and pain [8, 40, 53, 56, 58, 61, 63, 64, 80, 100, 105, 109].

Employment was often disrupted by menstruation, particularly for those experiencing menstrual disorders, including not going to work, leaving work, or having their quality of work impacted by the physical symptoms and/or behavioural expectations of menstruation, including hiding their menstrual status [8, 34, 40, 47, 53, 55–58, 61, 64, 71, 76, 80, 103, 105, 112, 114, 119, 122, 129]. Those who were able to be more flexible (e.g., those in more senior positions) managed their tasks and schedules to reduce the impact of menstruation on their employment [80, 105, 111, 112, 114, 119, 122]. Others went to work despite menstrual symptoms for fear of losing their job [53, 56, 61, 76, 80, 108, 112, 129].

Many participants abstained from physical activity, including swimming, during menstruation [8, 32, 34, 36, 38, 40, 41, 45, 47, 49–51, 54, 56, 60, 63, 65, 71, 72, 77, 80, 81, 87, 93–95, 99, 105, 108, 109, 112, 116, 127, 141]. In the early to mid-20th C this was often due to external behavioural expectations (or outright sanctions imposed by mothers) that they would not participate [36, 41, 45, 50, 56, 63, 65, 71, 93–95, 106, 108, 116] or a lack of appropriate menstrual materials for the particular activity, for example where tampons for swimming were not yet available, the participant chose not to use them, or the participant was not allowed to use them [32, 38, 56, 95, 109, 112]. In recent decades abstention from physical activity was more commonly due to study participants choosing not to engage due to physical menstrual symptoms [34, 40, 51, 77, 80, 87, 99, 127]. There were also participants who did partake in physical

activity during menstruation but worried that during this participation their menstrual status would be revealed [8, 27, 38, 49, 54, 56, 81, 106, 108].

There were often individuals who chose not, or were not allowed, to participate in certain activities. Those who reached menarche in the early to mid-20th C were often banned from taking part in activities that were considered 'cold' (e.g., bathing or sitting on cold surfaces) because it was considered dangerous to their health [41, 45, 50, 56, 63, 65, 71, 81, 84, 94, 95, 108]. Some individuals were not allowed to participate in religious practices whilst menstruating (within this review examples were given from those practicing Christianity, Judaism, Hinduism and Islam) [8, 60, 63, 67, 86, 92, 120], and some religions prohibited menstrual sex [67, 86, 105]. Where adolescent participants missed education during menstruation it was normally due to pain [8, 32, 34, 49, 56, 58, 61, 64, 65, 71, 80, 87, 99, 103, 122, 127, 129]. Many participants did not engage socially during menstruation [34, 36, 40, 41, 48, 58, 61, 71, 72, 80, 81, 87, 92, 94, 95, 99, 112, 119, 122, 129, 135]; some were not allowed to (normally by mothers) [36, 38, 41, 45, 65, 71, 72, 92–94, 119]. Those who reached menarche in the early-mid 20th C were often instructed not to interact with adolescent boys once menstruation began [36, 38, 65, 71, 81, 92, 93, 119].

**Relationships.** Menstruation often had an impact on participants' relationships. Impacts on intimate partner relationships were commonly linked to menstrual sex [33, 38, 47, 49, 56, 60, 61, 64, 73, 74, 81, 86, 93, 100, 104, 105, 112, 116, 119, 129, 140]; many chose not to engage due to low self-esteem and a concern that by wanting or having sex during menstruation their partner might consider them a "dirty cow" [pg. 86, 119] or "revolting" [pg. 31, 116], seemingly driven by internalised menstrual stigma.

The association between womanhood and menstruation sometimes led to closer relationships between cis-women and girls in their personal and professional lives, often corresponding to positive emotions such as happiness and pride [24, 32, 35, 38, 40, 43, 47, 49, 56, 59–61, 64, 68–71, 78, 88, 92–98, 105, 106, 109, 110, 113, 119, 120, 122, 135, 136, 140]. However, this association negatively impacted on the relationships of some non-binary and transgender people who menstruate, where it could lead to an "insider/outsider sort of thing where I experience this, but I'm not one of you" [pg. 381, 25].

## Discussion

The large number of studies of high or medium level trustworthiness and relevance enabled us to prepare an evidence synthesis and develop an integrated model which adequately captures the experiences of many of those who have menstruated in HICs over the past century, with some insights for specific sub-populations where multiple studies have been conducted. Across the timespan of studies and the multiple geographical contexts, the lived experiences of people who menstruate reflected consistent themes and relationships. Participants commonly expressed many negative experiences and detrimental impacts linked to menstruation–much more frequently than positive experiences and beneficial impacts. Although we should not conclude that the majority of those who menstruate in HICs are negatively affected, as often participants in these studies were recruited specifically to discuss negative experiences, it is clear that many people who menstruate within HICs have experienced negative wellbeing related to menstruation. The integrated model highlights particular themes and pathways which could be addressed in future to improve menstrual health.

### Some common pathways of menstrual experience in HICs

Socio-cultural context, particularly the stigmatisation of menstruation and gender norms related to managing menstruation, often manifested in behavioural expectations being placed

on those who menstruate, particularly to conceal menstrual status, often referred to as 'menstrual etiquette' [146]. Difficulties in abiding by expectations to contain menstrual fluid and conceal menstrual status often resulted in negative experiences, including distress and bother, as well as increased mental burden and consequences for participation and intimate relationships. Over the timespan of studies reviewed there was an increase in satisfaction with the menstrual materials on offer but concerns around adequately concealing menstrual status persisted.

Social support influenced the amount of knowledge participants had regarding the biology and practical management of menstruation. Where cis-women and girls felt they had adequate social support and knowledge, this sometimes led to happiness and improved relationships with other cis-women and girls, particularly at menarche. However, it was more common for participants to feel they received inadequate social support or knowledge about menstrual health and hygiene, which led to negative experiences, including shame and a lack of confidence to engage in activities, impacting participation and increasing mental burden. Knowledge of menstruation increased over time in the reviewed studies, reducing the negativity associated with menarche.

Resource limitations, particularly a lack of access to menstrual materials and facilities, were sometimes driven by the socio-cultural context itself, such as the lack of policy and public attention given to the menstrual health of low-income individuals or those who identify as non-binary or trans-men. Resource limitations contributed to negative experiences, including feelings of personal disgust at participants' own inability to manage their menstruation hygienically and in their preferred way. Such experiences often led to significant mental burden and a reduced participation in activities. The resource limitation theme and its implications were heavily informed by very recent, exploratory studies [10, 37, 48, 62, 65, 126, 132]. There is thus far insufficient evidence capturing the unmet menstrual health needs of marginalised and socioeconomically disadvantaged populations. Most studies focused on higher-income, adult groups and limited studies were identified responding to current policy priorities around inadequate access to products and supportive infrastructure for menstrual health, and adolescent menstrual health.

Individual menstrual factors such as pain, fatigue and gastrological and neurological symptoms were commonly associated with negative experiences, and led to increased mental burden, as well as detrimental impacts on participation and relationships. Participants who experienced irregular periods suffered particularly high mental burden from the constant need to be 'prepared' in case menstrual bleeding began unexpectedly. Those with menstrual disorders expended significant energy hiding their symptoms from employers and/or feeling guilty about letting colleagues and family members down. However, where healthcare workers were supportive and pain management effective, some participants did feel relief and reduced mental burden, and saw an improvement in their participation and relationships.

## Comparison to LMIC model

In both the HIC and LMIC bodies of evidence the socio-cultural context influenced behavioural expectations, impacting menstrual experiences and subsequent consequences for the lives of participants. The influence of menstrual stigma on menstrual experience and wellbeing was remarkably similar. As Hennegan *et al.* stated "Women and girls internalised menstrual restrictions and stigma and sought to regulate their behaviour accordingly. This impacted confidence to engage in other activities during menstruation and added to experiences of shame because a failure to hide menses was viewed as a personal failure to maintain feminine standards or menstrual etiquette." [pg. 22, 15]. This could be written verbatim with reference to

the HIC model. The power of social support sources including mothers, friends, and health-care workers to positively or negatively influence the experience of menstruation, emotional responses experienced and participation in daily life also echoed across both syntheses, as did the role of knowledge about menstruation and its management in supporting confidence, positive experiences and wellbeing.

Less emphasis in HIC studies was placed on resource deficits and the economic and physical environment than in LMICs. In LMICs, poverty and difficulty accessing resources for menstrual management were a significant focus of studies and a salient burden for participants. This difference likely reflects the relative resource and access to infrastructure across these participant groups, although it should be noted that in more recent HIC studies where these deficits have been explored, similar concerns to LMIC participants have been raised around accessing acceptable, reliable, and comfortable menstrual materials, along with supportive spaces for changing and disposing of them [10, 37, 48, 62, 65, 126, 132].

In contrast to the LMIC studies reviewed, few HIC studies described a lack of confidence to manage menstrual bleeding. This may reflect that the HIC review included primarily studies of adult women's experiences, compared to the focus on adolescent populations in LMICs, as well as the availability of supportive resources and infrastructure. However, varied confidence to engage in other activities during menstruation, and an enduring emphasis on concealment, were clearly reflected in both bodies of evidence and contributed to negative impacts on mental burden and participation.

Many studies included in the HIC review emphasised experiences of needing to endure discomfort or pain to maintain participation in work or other activities during menstruation. This appeared in contrast to studies from LMICs which more commonly highlighted consequences in terms of missed school or social participation. Notably, more recent studies from LMIC settings with adolescent girls and adult women have reported pressure to carry on with expected activities while concealing menstruation [147, 148]. It is likely that this experience is shaped by the evolving sociocultural context and experience at different ages.

Impacts of menstrual experiences on physical health, specifically the reproductive tract infections and irritation noted in LMICs, were not observed in the HIC literature. The absence of such discussions may be due to more advanced health infrastructure, and resource availability.

Differences between the broadly similar HIC and LMIC models must be interpreted in light of contextual differences as well as differences in the bodies of research reviewed, including study aims and participant recruitment. Studies included in the LMIC review tended towards a post-positivist epistemology, recruited low-income participants, and were designed with the intention of providing practical and policy recommendations related to menstrual health and resource deprivation. Studies from HICs tended to focus on in-depth, social constructivist investigations of menstrual experiences, without the intention of developing practical recommendations. We suggest that insights gained from comparing the two models and interrogating the assumptions shaping research and discourse in the different settings could strengthen global menstrual health and hygiene research, practice, and policy.

## Strengths and limitations

Our comprehensive searching strategy and efforts to identify relevant grey literature are a strength of this review. Qualitative studies of particular menstrual disorders directly related to menstrual bleeding were included, however we were limited in our analysis of how these specifically manifested in comparison to the general population [e.g., heavy menstrual bleeding, 149, endometriosis, 150]. In addition, for practical reasons and to enable a clear comparison to

the LMIC study, we limited our review to menstrual bleeding experiences, but recognise the importance of researchers examining individual experiences during other parts of the menstrual cycle.

One intention of this review was to contribute to current policy debates and actions aimed at addressing period poverty in HICs, but we were limited in the conclusions that could be drawn as, until very recently, most studies have focused on higher income populations. Studies tended to have been conducted in Europe or North America, although our requirement that records be available in English likely contributed to this limitation. Inclusion of studies in other languages would strengthen our model and its broader applicability.

## Implications and conclusions

Our integrated model is the first to map experiences of menstruation in HICs. The model can be used as a framework for understanding the factors to be considered when seeking to improve menstrual experiences and menstrual health. For example, the model suggests that approaches to reduce stigma, combat restrictive behavioural expectations and improve knowledge, social support and pain management may represent key levers for improving menstrual health. New research conducted in HICs can be informed by this work, with the model providing guidance on important themes, relationships and population groups for further exploration.

## Supporting information

**S1 Checklist. PRISMA checklist.**
(PDF)

**S1 Text. Organisational and personal websites searched.** Identified via members of the Menstrual Health Hub (https://mhhub.org/community/global-mh-registry/) and partners of Menstrual Hygiene Day (https://menstrualhygieneday.org/get-involved/partnership/). Searched in September 2019 and updated in November 2020.
(PDF)

**S2 Text. Menstrual health researchers contacted directly (October 2019 and November 2020).**
(PDF)

**S1 Table. Quality appraisal of included studies.** References in black boxes indicate they are from a study with multiple publications included in this review (publications from the same study are grouped together in Table 2). Quality is assessed at the publication level here and at the study level for analysis. Adapted from the EPPI-Centre Checklist detailed in Rees, R., Oliver, K., Woodman, J., Thomas, J. (2009) *Children's views about obesity, body size, shape and weight: A systematic review.* EPPI-Centre, London:UK [18].
(PDF)

## Acknowledgments

We would like to thank all of the menstrual health researchers and practitioners who suggested studies for inclusion in this review. We would also like to specifically thank Sophie Rowson for assisting us with the initial website search and AJ Lowik for assisting us with ensuring the manuscript uses gender inclusive language.

## Author Contributions

**Conceptualization:** Dani Jennifer Barrington, Emily Wilson, Julie Hennegan.

**Data curation:** Dani Jennifer Barrington, Hannah Jayne Robinson.

**Formal analysis:** Dani Jennifer Barrington, Emily Wilson, Julie Hennegan.

**Investigation:** Dani Jennifer Barrington, Hannah Jayne Robinson, Emily Wilson, Julie Hennegan.

**Methodology:** Dani Jennifer Barrington, Julie Hennegan.

**Project administration:** Dani Jennifer Barrington.

**Validation:** Emily Wilson, Julie Hennegan.

**Writing – original draft:** Dani Jennifer Barrington.

**Writing – review & editing:** Dani Jennifer Barrington, Hannah Jayne Robinson, Emily Wilson, Julie Hennegan.

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
