## [Decision Letter · Decision Letter 0]

11 Jun 2021

PONE-D-21-08777

Experiences of menstruation in high income countries: a systematic review, qualitative meta-synthesis and comparison to low- and middle-income countries

PLOS ONE

Dear Dr. Barrington,

Thank you for submitting your manuscript to PLOS ONE. After careful consideration, we feel that it has merit but does not fully meet PLOS ONE’s publication criteria as it currently stands. Therefore, we invite you to submit a revised version of the manuscript that addresses the points raised during the review process.

Please address issues raised during the view particularly around coding/developing themes.

We look forward to receiving your revised manuscript.

Kind regards,

Webster Mavhu

Academic Editor

PLOS ONE

Journal Requirements:

2. Please revise your PRISMA flow chart to ensure that you have included the reasons that full text articles were removed (listing how many were excluded for each reason).

3. Please ensure that you have stated all inclusion and exclusion criteria used to select manuscripts for inclusion.

4. We note that you use the word "meta-synthesis" in your title. As some readers may find this misleading as it is similar to "meta-analysis" please revise your title appropriately.

Additional Editor Comments (if provided):

This is a well-conducted and described study.

In addition to addressing issues raised by the reviewer, please:

a) ensure consistency in spelling meta-synthesis (sometimes metasynthesis)

b) minor edits upset, like (line 341); line 716 -However...

c) please check and confirm individuals acknowledged by name are happy with this

Reviewers' comments:

Reviewer's Responses to Questions

**Comments to the Author**

1. Is the manuscript technically sound, and do the data support the conclusions?

Reviewer #1: Yes

2. Has the statistical analysis been performed appropriately and rigorously? 

Reviewer #1: N/A

3. Have the authors made all data underlying the findings in their manuscript fully available?

Reviewer #1: Yes

4. Is the manuscript presented in an intelligible fashion and written in standard English?

Reviewer #1: Yes

5. Review Comments to the Author

Reviewer #1: This paper represents a thorough review of the literature and is poised to contribute greatly to the growing field of menstrual health. The authors review the qualitative literature on menstrual experiences in high-income countries and develop an integrated model that summarizes the major themes and relationships among them. The authors compare the findings to those from low- and middle-income countries and discuss how the integrated model differs across contexts. The model and review of the literature will be useful in guiding policy recommendations in high income contexts, as well as in highlighting areas in need of more rigorous research. The review is sound and the paper well written. I have just minor comments that the authors may consider to further improve the manuscript.

Abstract: The background provided is relevant, but it seems like a final sentence laying out more explicitly the aim of the present study would be helpful for readers before launching into the methods.

Data analysis:

• Authors discuss “coding studies”, but it is not clear exactly what data were coded. It would be helpful to have a quick mention of the process of extracting data from the published studies that was then used in the meta-synthesis. (e.g. Did the authors code any/all information that appeared in the “results” sections of published papers? Or did they also include information from “discussion” sections? Did they analyze author interpretations or only direct quotations from original studies, etc.)

• In step 4, two other authors coded 30% of the studies for validation purposes “without having viewed the coding template”. What type of coding were these authors conducting? Their approach is unclear—were they doing only inductive coding or did they also use the same framework approach as the first author (just without viewing what new codes were added by the first author)?

• In this section, it is unclear what “third-order constructs” is referring to in the analysis. Which are considered first and second-order constructs? This terminology does not come up again in the findings.

Table 2:

• The authors might consider changing the column “population size” to “number of participants” to more accurately describe the contents of that column

• Have the data collection methods of “focus group discussions” been combined/collapsed into the category of “group interviews”? Some disciplines distinguish between a group interview and focus group discussion (they are not considered the same method). In the table contents, it appears only the term “group interview” in used, yet in the footnote 2, the term “focus group” is employed. The authors might want to provide clarity around this.

• In the column “author stated analytical method,” some of the things listed are analysis methods but some are epistemological perspectives and/or theories or frameworks (social constructionism & symbolic interactionism) that guide methods but are not analysis “methods” in themselves.

Results:

• At the end of line 338, I think a word is missing after “regards”. Should this be “with regards to menarche”? (same issue in line 486, 554)

• In line 366, it is unclear whether the “media rhetoric that stigma around menstruation in HICs is decreasing” was a finding from the studies in the review. This may be more appropriate to discuss in the discussion section rather than the findings if not.

Discussion:

• Information in lines 791-797 appears to be new results that were not presented earlier in the findings section.

6. PLOS authors have the option to publish the peer review history of their article (what does this mean?). If published, this will include your full peer review and any attached files.

Reviewer #1: No

---

## [Author Response · Author response to Decision Letter 0]

27 Jun 2021

We have addressed the comments of the editor and reviewer in our Response to Reviewers document, uploaded.

---

## [Editor Report · Decision Letter 1]

8 Jul 2021

Experiences of menstruation in high income countries: a systematic review, qualitative evidence synthesis and comparison to low- and middle-income countries

PONE-D-21-08777R1

Dear Dr. Barrington,

We’re pleased to inform you that your manuscript has been judged scientifically suitable for publication and will be formally accepted for publication once it meets all outstanding technical requirements.

Kind regards,

Webster Mavhu

Academic Editor

PLOS ONE
---

## [Editor Report · Acceptance letter]

12 Jul 2021

PONE-D-21-08777R1 

Experiences of menstruation in high income countries: a systematic review, qualitative evidence synthesis and comparison to low- and middle-income countries 

Dear Dr. Barrington:

I'm pleased to inform you that your manuscript has been deemed suitable for publication in PLOS ONE. Congratulations! Your manuscript is now with our production department. 

Kind regards, 

on behalf of

Dr. Webster Mavhu 

Academic Editor

PLOS ONE